# Hyperbolic Visual-Semantic Alignment for Structural Visual Recognition

## Abstract

Visual and semantic concepts inherently organize themselves in a hierarchy, where a higher-level textual concept, *e.g.*, `Animal`, entails all images containing, *e.g.*, `Cat`. Despite being intuitive, conventional visual recognition systems strive to establish single-level correspondence between images and semantic concepts, and do not explicitly capture the hierarchical relationships that exist. We present HVSA to probe multi-level semantic information, from fine-grained to fully abstracted, within the tree-shaped hierarchy to realize structural visual recognition. Our main idea is to learn shared representations of images and semantic concepts in the hyperbolic space. Hyperbolic spaces possess suitable geometric properties to embed tree-like data structures, thus will help capture the underlying hierarchy. While it is challenging to acquire structure alignment of the two modalities, we achieve the goal through a joint optimization process guided by two primary objectives. First, we propose *hierarchy-agnostic visual-semantic alignment*, which leverages a Gaussian mixture VAE to establish a "flat" representation space shared by both modalities. Second, we introduce *hierarchy-aware semantic learning* to cultivate a "hierarchical" feature space for semantic concepts *solely* through hyperbolic metric learning. These two distinct objectives operate on different granularity and synergistically contribute to hierarchical alignment of visual-semantic features, ultimately enhancing structural image understanding. HVSA shows high efficacy and generality, as evidenced by its notable performance improvements across six datasets, for both image-level (*i.e.*, ImCLEF07A, ImCLEF07D and tieredImageNet-H) and pixel-level (*i.e.*, Cityscapes, LIP, and PASCAL-Person-Part) visual recognition. Our code shall be released.

## 1 Introduction

Hierarchical semantic concepts occur naturally and frequently in the real world. A huge spectrum of applications are characterized by hierarchical relationships between classes, ranging from text categorization (Dumais & Chen, 2000; Rousu et al., 2005), to functional genomics (Guan et al., 2008; Barutcuoglu et al., 2006). These problems share the common property that a class can be abstracted by more general classes at separate levels of a tree hierarchy.

As humans, we can easily organize semantic into a meaningful hierarchy, *e.g.*, `Animal → Dog → Hunting Dog` or `Animal → Okapi`. This inductive bias enables us to reasonably interpret, *e.g.*, images of `Okapi`, as belonging to the broader category of `Animal`, since we, non-mammal experts, might have no any sense of how a rare animal `Okapi` looks like. Hierarchical representations can not only properly handle such open-world cases, but also show the potential to improve interpretability (Nauta et al., 2021) and enable better exploratory data analysis of large datasets (Deng et al., 2009).

Many efforts have been devoted to accommodate the underlying hierarchical taxonomy into deep models for structural visual recognition, to yield structured predictions that conform to the taxonomy. One line of work imposes the inherent logical constraints of concepts to the losses of neural networks (Giunchiglia & Lukasiewicz, 2020; Li et al., 2020; Wehrmann et al., 2018; Chen et al., 2022), which can enhance prediction accuracy but doesn't guarantee consistency in predictions, especially during inference. Another set of methods (Bi & Kwok, 2011; Chen et al., 2020; Desai et al., 2023) aims to represent labels as low-dimensional vectors. While vector-based approaches offer interpretability, they are limited because these embeddings only capture correlations between labels and don't effectively

learn the hierarchical structure among labels. Furthermore, methods adapting network architectures (Ahmed et al., 2016; Wang et al., 2021) to accommodate hierarchical multi-label recognition tasks show promising results with a significant cost to the model's generalization capability.

In this work, we propose a novel approach called Hyperbolic Visual-Semantic Alignment (HᴠSA). In our task setting, labels are organized hierarchically based on a given label taxonomy rather than being mutually exclusive. Benefiting from hyperbolic wrapped normal (Mathieu et al., 2019; Nagano et al., 2019), we model more explicitly visual-semantic interactions, with the hierarchical structure shaped by hyperbolic geometrics. We assume that these embeddings are generated from a shared multimodal latent space for images and labels. We determine the optimal configuration of the multimodal latent space with two insights: (1) each visual entity (*e.g.*, image, pixel) is represented through a composition of multiple semantic labels, and (2) these labels are correlated in nature, collectively forming a tree-shaped hierarchy. To address (1), we use KL divergence to align the latent distributions of images and labels, which leads to a *hierarchy-agnostic visual-semantic alignment* component for multimodal alignment. Furthermore, to address (2), we propose *hierarchy-aware semantic learning* to interpret the hierarchical structure of semantic concepts defined in the label taxonomy. Unlike previous approaches that focus on complex network designs, we start from the perspective of embedding space. We believe that a pair of labels with superclass-subclass relationships have similar or even overlapping embeddings. Our method uses hyperbolic entailment cones to measure distances between categories in the embedding space. Through hyperbolic metric learning driven by the given taxonomy, HᴠSA is able to build a highly hierarchical feature space for semantic concepts.

Using the label hierarchy to guide the classification models, we are able to bridge one gap in the way machines and humans deal with visual understanding. Extensive experiments (§5) on six datasets verify the generalization and effectiveness of HᴠSA.

## 2 RELATED WORK

**Hierarchical Visual Recognition.** Considering how to learn the hierarchy of categories is a common challenge across various machine learning application domains, including functional genomics (Giunchiglia & Lukasiewicz, 2020), multi-label image classification (Bengio et al., 2010; He et al., 2021), and hierarchical semantic segmentation. The focus lies in ensuring that the learned knowledge about hierarchy aligns with the label taxonomy. As a result, a series of algorithms has been proposed: i) Previous work (Bertinetto et al., 2020; Bilal et al., 2017; Giunchiglia & Lukasiewicz, 2020) enforces hierarchical constraints by encoding label hierarchy in loss functions to ensure consistency between prediction results and the class hierarchy. ii) Adapting the classifier architecture to accommodate labels with hierarchical structures is effective (Ahmed et al., 2016; Cerri et al., 2014; 2016; Wang et al., 2021). Their hierarchies are typically fixed and tailored for specific downstream tasks such as classification, which limit their ability to generalize across tasks. iii) Some human parsers (Liang et al., 2018b; Wang et al., 2019; 2020b; Zhu et al., 2018; Zhou et al., 2021) attempt to explore human hierarchical relations, and certain methods add structured knowledge to semantic segmentation networks. With the exploration of representation learning in embedding spaces, Learning a shared latent space for features and labels is a common and useful idea. Methods adopting this idea typically include modules that directly map multi-hot labels to embeddings (Yeh et al., 2017; Chen et al., 2019). However, these approaches overlook the implicit hierarchical relationships between semantic labels. In addition, HSSN (Li et al., 2022) proposes a general framework for both HSS network design and training by leveraging pixel-level hierarchical reasoning and representation learning. We aim to truly convey the hierarchical relationships between labels to the model by learning hierarchical representations of the labels themselves.

**Hyperbolic Representations Learning.** Traditionally, representations are learned in Euclidean space. However, hyperbolic space representation learning has gained recognition in the deep learning literature for representing tree-like structures and taxonomies (Ganea et al., 2018; Law et al., 2019; Nickel & Kiela, 2017; Sala et al., 2018), text (Tao et al., 2020), and graphs (Lou et al., 2020). The datasets often exhibit a hierarchical structure, motivated by two key factors that make hyperbolic representations learning a suitable choice. First, generality: the hypernym-hyponym relationship is a natural feature of words, exemplified by WordNet (Miller et al., 1990). Hyperbolic representations learning is widely utilized for learning word and image embeddings while preserving this property (Ganea et al., 2018; Liu et al., 2020). Second, compositionality: hierarchies often emerge from

the composition of basic elements. This observation has driven prior work to apply hierarchical representations learned in hyperbolic space such as image classification (Khrulkov et al., 2020), segmentation(Weng et al., 2021) and action recognition(Long et al., 2020). In this paper, our primary focus is to obtain a hierarchical space through representation learning that can capture semantic structures, and achieve structure alignment between the two modalities.

## 3 PRELIMINARIES OF HYPERBOLIC LEARNING

**Hyperbolic Geometry & Poincaré Embeddings.** Hyperbolic geometry is a non-Euclidean geometry with constant negative sectional curvature. A pivotal property of hyperbolic space is its exponential volume expansion concerning a ball with radius r, in contrast to the polynomial growth exhibited in Euclidean space. This inherent exponential growth characteristic plays a pivotal role in substantiating the natural aptitude of hyperbolic embeddings for capturing hierarchical structures. There exist multiple, equivalent models for hyperbolic space, and we base our approach on the Poincaré ball model, due to its conformality and convenient parameterization. The Poincaré ball model is the Riemannian manifold $\mathcal{P}^d = (\mathbb{B}^d, g_p)$, where $\mathbb{B}^d = \{\boldsymbol{x} \in \mathbb{R}^d : \|\boldsymbol{x}\| < 1\}$ is the open $d$-dimensional unit ball equipped with the Riemannian metric tensor $g_p$ and metric distance $d_p$:

$$g_p(\boldsymbol{x}) = 4(1 - \|\boldsymbol{x}\|^2)^{-2}g_e, \qquad d_p(\boldsymbol{x}, \boldsymbol{y}) = \cosh^{-1}\left(1 + 2\frac{\|\boldsymbol{x} - \boldsymbol{y}\|^2}{(1 - \|\boldsymbol{x}\|^2)(1 - \|\boldsymbol{y}\|^2)}\right), \quad (1)$$

where $g_e$ is the Euclidean metric metric.

To be able to operate on the Poincaré ball, we use the exponential mapping, $i.e.$, $\exp_{\boldsymbol{x}} : \mathbb{R}^d \to \mathbb{B}^d$, to map from Euclidean space to Poincaré ball. We can also use the logarithmic mapping, $i.e.$, $\log_{\boldsymbol{x}} : \mathbb{B}^d \to \mathbb{R}^d$, to reverse this process. Their closed-form expressions are defined as:

$$\exp_{\boldsymbol{x}}(\boldsymbol{v}) = \boldsymbol{x} \oplus \left(\tanh\left(\frac{\lambda_{\boldsymbol{x}}\|\boldsymbol{v}\|}{2}\right)\frac{\boldsymbol{v}}{\|\boldsymbol{v}\|}\right), \qquad \log_{\boldsymbol{x}}(\boldsymbol{u}) = \frac{2}{\lambda_x}\tanh^{-1}\left(\frac{\|\boldsymbol{u} - \boldsymbol{x}\|}{\sqrt{1 - \|\boldsymbol{x}\|^2}}\right) \cdot \frac{\boldsymbol{u} - \boldsymbol{x}}{\|\boldsymbol{u} - \boldsymbol{x}\|} \quad (2)$$

where $\|.\|$, $\oplus$ denotes the euclidean norm and Möbius addition (Ungar, 2008) respectively. In practice, the base point $\boldsymbol{x}$ is typically set to $0$ which has been found to have minimal impact on results while simplifying the associated formulas.

**Wrapped Normal.** In order to parametrise distributions on the Poincaré ball, we consider one canonical generlization of normal distributions to Riemannian manifolds, which is called *wrapped normal distribution* (Mathieu et al., 2019; Nagano et al., 2019). It is defined on an arbitrary Riemannian manifold as the push-forward measure obtained by mapping a normal distribution along the exponential map $\exp_i$. On the Poincaré ball, the probability density function (PDF) of the wrapped normal with mean $\boldsymbol{\mu}$ and covariance $\boldsymbol{\Sigma}$ is given by:

$$\mathcal{N}_p(\boldsymbol{z}|\boldsymbol{\mu}, \boldsymbol{\Sigma}) = \mathcal{N}(g_p(\boldsymbol{\mu})|0, \Sigma)\left(\frac{d_p(\boldsymbol{\mu}, \boldsymbol{z})}{\sinh(d_p(\boldsymbol{u}, \boldsymbol{z}))}\right), \tag{3}$$

where $\mathcal{N}$ refers to the normal distribution in Euclidean space.

## 4 OUR APPROACH: HYPERBOLIC VISUAL-SEMANTIC ALIGNMENT

As shown in Fig. 1, HVSA consists of two parts: *hierarchy-agnostic visual-semantic alignment*(§4.1) and *hierarchy-aware semantic learning*(§4.2). We tackle the task of hierarchical multi-label classification in which given a label taxonomy $\mathcal{G}$, a neural network $\mathcal{F}(\cdot; \theta)$ must learn to associate an image $X \in \mathbb{R}^{h \times w \times 3}$ to $L$ interdependent labels $\boldsymbol{y} \in \{0, 1\}^L$ from the taxonomy. We describe a set of labels $\{y\}$ that have hierarchical relationships in the dataset using the tree-like hierarchical structure $\mathcal{T} = (V, E) \subseteq \mathcal{G}$ (Li et al., 2022), which consists of a set of nodes $V = \{v_1, v_2, ..., v_n\}$ denoting semantic classes and an undirected edge $E$ between nodes with semantic relations. An undirected edge $(v_i, v_j) \in E$ indicates that the class $j$ is a superclass of label $i$, $e.g.$, the relationship between vehicle and car can be expressed as (vehicle, car). For nodes $v_i, v_j$, the semantic similarity between $v_i$ and $v_j$ can be measured by computing the distance $D(v_i, v_j)$ in $\mathcal{T}$. $D(v_i, v_j)$ is defined as the shortest path between $v_i$ and $v_j$ in $\mathcal{T}$ and reflects the semantic proximity between them.

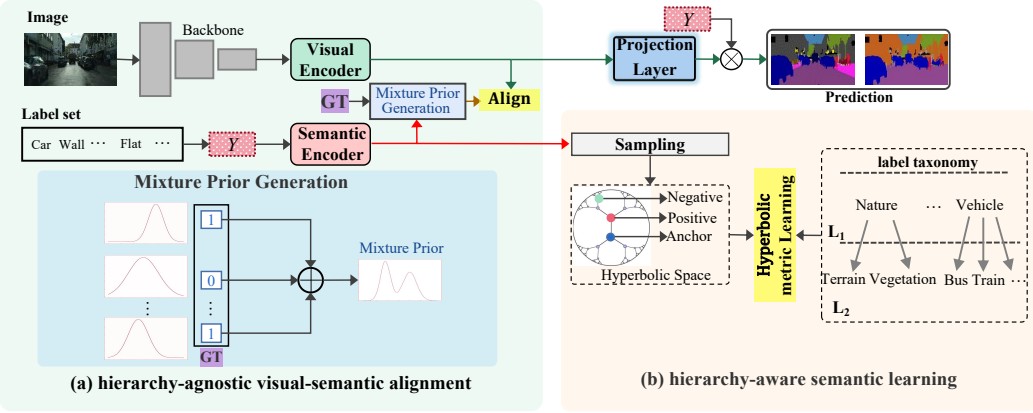

Figure 1: HVSA framework consists of two parts: given a feature and a set of labels, visual feature is aligned with mixed label distributions through *hierarchy-agnostic visual-semantic alignment* (§4.1). Simultaneously, to enhance hierarchical knowledge, *hierarchy-aware semantic learning* (§4.2) is employed to assist semantic concepts in hyperbolic space in adhering to hierarchical knowledge.

## 4.1 HIERARCHY-AGNOSTIC VISUAL-SEMANTIC ALIGNMENT

**Hyperbolic Probabilistic Label Embedding.** Unlike previous works (Long et al., 2020; Liu et al., 2020; Ghadimi Atigh et al., 2021) that embed label hierarchies into hyperbolic prototypes, we model each label in a probabilistic manner using the wrapped normal distribution. For each label $l$ (*e.g.*, *person*), we map its randomly-initialized label embedding $\boldsymbol{f}_l \in \mathbb{R}^d$ to a wrapped normal distribution $\mathcal{N}_p(\boldsymbol{\mu}_l, \boldsymbol{\Sigma}_l)$. Here $\boldsymbol{\Sigma}_l = \mathrm{diag}(\boldsymbol{\sigma}_l^2)$, and $\boldsymbol{\mu}_l \in \mathbb{R}^d$ and $\boldsymbol{\sigma}_l \in \mathbb{R}^d$ are derived from a semantic encoder (see §4.3). In this manner, we represent each label as a unimodal Gaussian in the hyperbolic space, and for each sample with multiple labels, its label embedding belongs to a Gaussian mixture subspace. Formally, for a random variable $\boldsymbol{z}$, its probability density function (PDF) is given as:

$$p_\theta(\boldsymbol{z}) = \frac{1}{\sum_{l=1}^L y_l} \sum_{l=1}^L \mathbb{1}(y_l = 1)\mathcal{N}_p(\boldsymbol{z}|\boldsymbol{\mu}_l, \boldsymbol{\Sigma}_l), \tag{4}$$

where $\mathbb{1}(y_l = 1)$ refers to an indicator function.

**Hyperbolic Visual Embedding.** Next, we learn a projection of an input image $x \in \mathbb{R}^{h \times w \times 3}$ to a hyperbolic manifold so that we are able to compute proximity to label embedding. We achieve this through a standard feature extractor $h(\cdot; \theta)$ (*e.g.*, ResNet (He et al., 2016)) with parameters $\theta$, followed by a visual encoder with an exponential map (Eq. 2) from the tangent space $\mathcal{T}_x\mathcal{M}$ to a hyperbolic manifold $\mathcal{M}$:

$$\hat{\boldsymbol{x}} = \exp_{\boldsymbol{v}}(\boldsymbol{x}), \qquad \boldsymbol{x} = h(x; \theta). \tag{5}$$

Here $\boldsymbol{x}$ is the representation of image $x$ in Euclidean space, which are projected to the the hyperbolic space, yielding $\hat{\boldsymbol{x}}$.

**Hyperbolic Gaussian Mixture VAE.** Last, we seek to align visual embedding with label embedding to yield a shared representation space. This can be achieved based on VAE, which uses variational inference and probabilistic modeling to learn the latent representations of data. Most VAE-based frameworks optimize over an evidence lower bound (ELBO) (Doersch, 2016):

$$\mathrm{ELBO} = \int_{\mathcal{M}} \ln \frac{p_\theta(\boldsymbol{x}|\boldsymbol{z})p(\boldsymbol{z})}{q_\phi(\boldsymbol{z}|\boldsymbol{x})} q_\phi(\boldsymbol{z}|\boldsymbol{x})d\mathcal{M}(\boldsymbol{z}) = \mathbb{E}_{\boldsymbol{z} \sim q_\phi(\cdot|\boldsymbol{x})\mathcal{M}(\cdot)}[\ln p_\theta(\boldsymbol{x}|\boldsymbol{z}) - D_{\mathrm{KL}}(q_\phi(\boldsymbol{z}|\boldsymbol{x})||p(\boldsymbol{z}))], \tag{6}$$

Here $q_\phi(\boldsymbol{z}|\boldsymbol{x})$ represents the variational posterior distribution that is an approximation to the intractable true posterior $p_\theta(\boldsymbol{z}|\boldsymbol{x})$. We use a feature extractor to obtain a standard posterior and match it with the mixed prior distribution. However, unlike traditional VAEs, we cannot compute the KL divergence analytically in this context. Inspired by (Shu, 2016), we approximate the KL term by:

$$D_{\mathrm{KL}}(q_\phi(\boldsymbol{z}|\boldsymbol{x})||p(\boldsymbol{z}))] \approx \ln p(\boldsymbol{z}^k) - \ln q_\phi(\boldsymbol{z}^k|\boldsymbol{x}) \tag{7}$$

where $\boldsymbol{z}^k \sim q_\phi(\boldsymbol{z}|\boldsymbol{x})\sqrt{|G(\cdot)|}$ and $G$ denotes the Riemannian metric. The reconstruction loss (Kingma & Welling, 2013) is a standard negative log-likelihood with decoder parameters $\varphi$:

$$\mathcal{L}_{\mathrm{RECON}} = -\mathbb{E}_{\boldsymbol{z}^k \sim q_\phi(\boldsymbol{z}|\boldsymbol{x})\sqrt{|G(\cdot)|}}[\ln p_\varphi(\boldsymbol{x}|\boldsymbol{z}^k)] \tag{8}$$

### 4.2 HIERARCHY-AWARE SEMANTIC LEARNING

§4.1 addresses the compositional properties of semantic concepts *solely*, but the learnt embedding space cannot be guaranteed to obey the semantic hierarchy. HVSA tackles this through hierarchy-aware semantic learning, which performs hyperbolic metric learning to reshape the semantic embedding space based on the hierarchical structure defined in label taxonomy.

**Hyperbolic Metric Learning.** Our method is based on a max-margin loss computed in hyperbolic space. Formally, denote $\{v_a, v_p, v_n\}$ as a triplet, where $v_a$, $v_p$, $v_n$ refer to anchor, positive and negative nodes (*i.e.*, categories) in $\mathcal{T}$, respectively. In our design, classes with higher semantic similarities (closer in the tree $\mathcal{T}$) to anchor are selected as positive samples, and vice versa. This means that we have $D(v_a, v_p) < D(v_a, v_n)$. Notably, our method is different from the standard triplet loss in which positive samples come from a same class as the anchor. With a triplet $\{v_a, v_p, v_n\}$, we formulate hyperbolic metric learning as the following max-margin loss:

$$\mathcal{L}_{\text{HM}} = \sum_{(\boldsymbol{z}_a, \boldsymbol{z}_p) \in \mathcal{P}} E(\boldsymbol{z}_a, \boldsymbol{z}_p) + \sum_{(\boldsymbol{z}_a, \boldsymbol{z}_n) \in \mathcal{N}} \max(0, \gamma - E(\boldsymbol{z}_a, \boldsymbol{z}_n)), \tag{9}$$

where $\boldsymbol{z}_a, \boldsymbol{z}_p, \boldsymbol{z}_n$ are feature embeddings of $v_a$, $v_p$, $v_n$, respectively. $\mathcal{P}$ and $\mathcal{N}$ represent sets of anchor-positive and anchor-negative pairs, respectively. $\gamma > 0$ is a margin. The energy $E(\boldsymbol{z}_a, \boldsymbol{z}_n)$ measures the penalty of a wrongly classified pair $(v_a, v_p)$, which in our case is computed as the minimum angle required to rotate the axis of the cone at $\boldsymbol{z}_a$ to bring $\boldsymbol{z}_p$ into the cone:

$$E(\boldsymbol{z}_a, \boldsymbol{z}_p) = \max(0, \Xi(\boldsymbol{z}_a, \boldsymbol{z}_p) - \psi(\boldsymbol{z}_a)). \tag{10}$$

The aperture of the cone is $\psi(\boldsymbol{z}_a) = \arcsin(K(1 - ||\boldsymbol{z}_a||^2)/||\boldsymbol{z}_a||)$ and $K$ is a hyper-parameter. $\Xi(x, y)$ computes the minimum angle between the axis of the cone at $x$ and the vector $y$:

$$\Xi(x, y) = \arccos\left(\frac{\langle x, y \rangle (1 + ||x||^2) - ||x||^2(1 + ||y||^2)}{||x|| ||x - y|| \sqrt{1 + ||x||^2 ||y||^2 - 2\langle x, y \rangle}}\right) \tag{11}$$

**Constructing Samples for Metric Learning.** During metric learning, we randomly sample a set of class triplets from $\mathcal{T}$ in the form $\{v_a, v_p, v_n\}$, where $D(v_a, v_p) < D(v_a, v_n)$. Then, for each class, say $v_a$, we obtain its feature embedding $\boldsymbol{z}_a$ by sampling from corresponding Gaussian distribution defined in §4.1. To maintain a continuous gradient during the sampling process, we apply the reparameterization trick (Kingma & Welling, 2013; Mathieu et al., 2019). We first sample a random variable $\epsilon \sim \mathcal{N}(0, 1)$, and then obtain the embedding by: $\boldsymbol{z}_a = \boldsymbol{\mu}_{v_a} + \epsilon \boldsymbol{\sigma}_{v_a}$.

### 4.3 DETAILED NETWORK ARCHITECTURE

**Network Structure.** The visual encoder is an MLP with 3 hidden layers of sizes [512, 512, 256], while the semantic encoder has 2 hidden layers of sizes [512, 256]. $d$ is set to 512 by default.

**Model Prediction.** To compute predictions, the hyperbolic visual feature of each input image is first transformed into Euclidean space via a logarithm mapping layer (*c.f.* Eq. 2). Then, the feature is projected into a 512-dimensional space through a linear projection layer. Finally, the prediction score of each category is determined by the inner product between the visual feature and the label embedding of corresponding category.

**Overall Loss.** Our training loss $\mathcal{L}$ is a combination of a classification loss $\mathcal{L}_{\text{CLS}}$, the KL divergence $D_{\text{KL}}$ (Eq. 7), the reconstruction loss $\mathcal{L}_{\text{RECON}}$ (Eq. 8), as well as $\mathcal{L}_{\text{HM}}$ for hyperbolic metric learning (Eq. 9) for structural visual recognition, we combine them as follows:

$$\mathcal{L} = \mathcal{L}_{\text{CLS}} + \alpha \mathcal{L}_{\text{HM}} + \beta(D_{\text{KL}} + \mathcal{L}_{\text{RECON}}), \tag{12}$$

where the coefficients are empirically set as $\alpha = 0.2$ and $\beta = 4$. Concretely, for image classification tasks, we use cross-entropy loss as $\mathcal{L}_{\text{CLS}}$, while for segmentation, focal loss (Lin et al., 2017) is used.

## 5 EXPERIMENT

### 5.1 EXPERIMENTAL SETUP

**Datasets.** We evaluate HVSA on total six standard benchmark datasets (§B.1). ImCLEF07A and ImCLEF07D (Dimitrovski et al., 2011) are for validating HVSA in hierarchical classification tasks.

Table 1: Comparison of performance and consistency on ImCLEF07A and ImCLEF07D.

| Dataset | Metric | HvsA | BoxE (Abboud et al., 2020) | MVM (Wang et al., 2018) | MHM (Chen et al., 2020) | MBM (Patel et al., 2022) |
|---------|--------|------|------|------|------|------|
| ImCLEF07A | MAP | **92.21** | 83.71 | 77.14 | 65.29 | 91.45 |
| | CMAP | **92.44** | 84.73 | 76.56 | 66.01 | 91.73 |
| | CV | **3.18** | 12.73 | 23.02 | 4.75 | 5.65 |
| ImCLEF07D | MAP | **90.61** | 87.95 | 88.49 | 75.72 | 89.49 |
| | CMAP | **91.42** | 88.93 | 86.89 | 76.98 | 89.99 |
| | CV | **5.71** | 11.93 | 10.72 | 7.52 | 7.16 |

A large dataset TieredImageNet-H (Bertinetto et al., 2020) containing 608 categories is used to verify that the exponential growth of the capacity of the hyperbolic space is conducive to the hierarchical classification of large-scale labels. Additionally Cityscapes (Cordts et al., 2016), a commonly used urban scene analysis dataset, is used to demonstrate the performance of our model in pixel-level classification tasks. The LIP (Liang et al., 2018a) and PASCAL-Person-Part (Xia et al., 2017) datasets are employed to verify the efficacy of HvsA in human parsing task, and in cases of small-scale data.

**Implementation Details.** Following MBM (Patel et al., 2022), we trained for 600 epochs on ImCLEF07A and ImCLEF07D using Adam as the optimizer with learning rate 1e-4 and batch size 8. For TieredImageNet-H, we adopt ResNet-18 pretrained on the ImageNet, using the SGD optimizer with learning rate 0.01 and batch size 4. Additionally, the images were cropped to 224x224 pixels, and the training was conducted for 100k iterations. For Cityscapes, LIP and PASCAL-Person-Part, we set hyper-parameters for training, following (Wang et al., 2019; Zhao et al., 2017; Zhang et al., 2020a; Li et al., 2022). All backbones are initialized by pre-trained parameters on ImageNet-1K (Deng et al., 2009) and the remaining layers are randomly initialized. For data preparation, following (Ruan et al., 2019; Wang et al., 2019), we use standard data augmentation techniques, flipping in horizon and random scaling with a factor in [0.5, 2.0]. We adopt the standard SGD solver as the optimizer with a momentum 0.9 and weight decay of $1e^{-4}$ for segmentation and $5e^{-4}$ for classification. In addition, we train 80K iterations for Cityscapes, with a batch size of 8 and a training crop size of 512×1024. For PASCAL-Person-Part and LIP, we respectively train models for 80K and 120K iterations with batch size 16 and crop size 480×480. In the initial training stages, the shared embedding space quality is suboptimal, and large learning rate might lead to model divergence. Therefore, we begin with a warm-up phase. Furthermore, our learning rate follows the cosine annealing policy (Loshchilov & Hutter, 2016) with initial value of 2e-4 and range is [0, 1e-3].

**Metrics.** For image classification, we employ the Mean Average Precision (MAP) as the primary metric and additionally report Constraint Violation (CV) (Patel et al., 2022) and Mean Average Precision post Coherence correction (CMAP) (Patel et al., 2022) on ImCLEF07A and ImCLEF07D. As conventions, we use top-1 error on tieredImageNet-H, and the mIoU for semantic segmentation. Following (Li et al., 2022), we also report $mIoU^l$, the average score in each hierarchy level $l$ on Cityscapes, LIP and PASCAL-Person-Part. Please see §B.2 for more details.

## 5.2 COMPARISON WITH STATE-OF-THE-ART METHODS

**ImCLEF07A and ImCLEF07D.** In Table 1, we compare HvsA against three competitors on ImCLEF07A and ImCLEF07D. Here MBM (Patel et al., 2022) is the current state-of-the-art that utilizes the geometry and probabilistic semantics of box embeddings to model label-label interactions in multi-label classification. The results show that HvsA yields a promising gain against MBM. Particularly, HvsA solidly outperforms MBM in terms of CV, *i.e.*, **3.18%** *vs.* 5.65% on ImCLEF07A and **5.71%** *vs.* 7.16% on ImCLEF07D. This reveals that by addressing multimodal feature alignment in hyperbolic space, the feature space derived from our model more closely adhere to label taxonomy.

**TieredImageNet-H.** HvsA also shows significant improvements in hierarchical classification of natural images, as shown in Table 2. HvsA consistently outperforms various methods, achieving a remarkable reduction in error rate by 1.37% compared to the previous SOTA method, SOFT-LABELS (Bertinetto et al., 2020). Compared to medical images, this dataset contains a large number of labels, which poses demands on the capacity of the representation space to achieve good performance. In contrast to the polynomial-level growth of traditional Euclidean spaces, hyperbolic spaces exhibit

Table 2: Results on the tieredImageNet (Bertinetto et al., 2020).

| Method | Top-1 error |
|---|---|
| BARZ&DENZLER (Barz & Denzler, 2019) | 39.03 |
| DEVISE (Frome et al., 2013) | 31.69 |
| YOLO-v2 (Redmon & Farhadi, 2017) | 30.43 |
| HXE (Bertinetto et al., 2020) | 27.68 |
| SOFT-LABELS (Bertinetto et al., 2020) | 27.78 |
| **HvsA (Ours)** | **26.41** |

Table 3: Per-hierarchy comparison of mIoU on Cityscapes (Cordts et al., 2016) `val`. $*$ and $+$ denote using DeepLabV3+ (Chen et al., 2018) and OCRNet (Yuan et al., 2020) as segmentation head, respectively.

| Method | Year | Backbone | $mIoU^1$ | $mIoU^2$ |
|---|---|---|---|---|
| DeepLabV2 (Chen et al., 2017) | CVPR17 | ResNet-101 | 70.22 | — |
| PSPNet (Zhao et al., 2017) | CVPR17 | ResNet-101 | 80.91 | — |
| PSANet (Zhao et al., 2018) | ECCV18 | ResNet-101 | 80.96 | — |
| PAN (Li et al., 2018) | ArXiv18 | ResNet-101 | 81.12 | — |
| DeepLabV3+ (Chen et al., 2018) | ECCV18 | ResNet-101 | 82.08 | 92.16 |
| DANet (Fu et al., 2019) | CVPR19 | ResNet-101 | 81.52 | — |
| Acfnet (Zhang et al., 2019) | ICCV19 | ResNet-101 | 81.60 | — |
| CCNet (Huang et al., 2019) | ICCV19 | ResNet-101 | 81.08 | — |
| HANet (Choi et al., 2020) | CVPR20 | ResNet-101 | 81.82 | — |
| HSSN$^*$ (Li et al., 2022) | CVPR22 | ResNet-101 | 83.02 | 93.31 |
| HRNet (Wang et al., 2020a) | TPAMI20 | HRNet-W48 | 81.96 | 92.12 |
| OCRNet (Yuan et al., 2020) | ECCV20 | HRNet-W48 | 82.33 | 92.57 |
| HSSN$^+$ (Li et al., 2022) | CVPR22 | HRNet-W48 | 83.37 | 93.92 |
| HvsA $^*$ | — | ResNet-101 | 84.31 | 93.97 |
| HvsA $^+$ | — | HRNet-W48 | **84.63** | **94.27** |

exponential-level growth in spatial capacity, which enables better performance in datasets with a large number of labels.

**Cityscapes.** Table 3 compares our method against twelve famous methods on Cityscapes `val`. Despite that the dataset has a relatively simple semantic hierarchy, the evaluation results demonstrate that our method achieves **84.31%/93.97%** and **84.63%/94.27%** at two different evaluation levels over the DeepLabV3+ (Chen et al., 2018) and OCRNet (Yuan et al., 2020), respectively. It performs consistently better than the previous SOTA, *i.e.*, HSSN$^+$, that addresses hierarchical learning in Euclidean space. In addition, we highlight that HvsA is superior to HSSN$^+$ in that it does not rely on label taxonomy during inference. The results also confirm the strong generalizability of HvsA.

**LIP.** The quantitative comparison results with sixteen methods on LIP `val` are summarized in Table 4. As seen, our approach consistently produces the best performance (**61.94%/94.95%/98.79%/**) across the three levels. Particularly, it outperforms HSSN (Li et al., 2022) by **1.5%** in terms of $mIoU^1$.

**PASCAL-Person-Part.** Low data volume should pose a challenge for VAE to generate shared representation spaces for two modalities. An interesting observation is that HvsA demonstrates superior performance even on small datasets. Table 5 provides an evaluation on PASCAL-Person-Part `test`, demonstrating the superior performance of our model (**76.37%/88.94%/97.88%**). Notably, our method outperforms the previous state-of-the-art method HSSN (Li et al., 2022) by **0.93%/0.74%/0.19%** in terms of $mIoU^1$ $mIoU^2$ and $mIoU^3$, respectively. These results serve as strong evidence of the efficacy of our approach in learning shared representation spaces.

## 5.3 ABLATION STUDY

**Analysis of Key Component.** Table 6 summaries the comparative results between our full model and ablated versions without specific key component. In the first row, a hierarchy-agnostic method is shown and trained by focal loss $\mathcal{L}_{FL}$. The second variant is training by a Gaussian mixture VAE without hyperbolic metric learning loss $\mathcal{L}_{HM}$. From the results, we can see that by introducing $D_{KL}$, we obtain consistent performance improvements in the four datasets across all the metrics. This confirms the efficacy of hierarchy-agnostic visual-semantic alignment, which can generalize well for image or pixel classification tasks. By further introducing hyperbolic metric learning loss

Table 4: Per-hierarchy comparison of mIoU on LIP (Liang et al., 2018a) `val`.

| Method | Year | Backbone | $mIoU^1$ | $mIoU^2$ | $mIoU^3$ |
|---|---|---|---|---|---|
| SegNet (Badrinarayanan et al., 2017) | TPAMI17 | ResNet-101 | 18.17 | — | — |
| FCN-8s (Long et al., 2015) | CVPR15 | ResNet-101 | 28.29 | — | — |
| DeepLabV2 (Chen et al., 2017) | CVPR17 | ResNet-101 | 41.64 | — | — |
| Attention (Chen et al., 2016) | CVPR16 | ResNet-101 | 42.92 | — | — |
| MMAN (Luo et al., 2018) | ECCV18 | ResNet-101 | 46.93 | — | — |
| DeepLabV3+ (Chen et al., 2018) | ECCV18 | ResNet-101 | 52.28 | 83.97 | 88.13 |
| CE2P (Ruan et al., 2019) | AAAI19 | ResNet-101 | 53.10 | — | — |
| BraidNet (Liu et al., 2019) | ACMMM19 | ResNet-101 | 54.42 | — | — |
| SemaTree (Ji et al., 2020) | ECCV20 | ResNet-101 | 54.73 | 87.12 | 90.78 |
| BGNet (Zhang et al., 2020a) | ECCV20 | ResNet-101 | 56.82 | — | — |
| PCNet (Zhang et al., 2020b) | CVPR20 | ResNet-101 | 57.03 | — | — |
| CNIF (Wang et al., 2019) | ICCV19 | ResNet-101 | 57.74 | 91.83 | 95.92 |
| HHP (Wang et al., 2020b) | CVPR20 | ResNet-101 | 59.25 | 93.43 | 97.41 |
| HSSN (Li et al., 2022) | CVPR22 | ResNet-101 | 60.37 | 94.75 | 98.86 |
| HRNet (Wang et al., 2020a) | TPAMI20 | HRNet-W48 | 57.23 | 91.21 | 95.53 |
| OCRNet (Yuan et al., 2020) | ECCV20 | HRNet-W48 | 58.47 | 92.56 | 96.78 |
| HvsA | — | ResNet-101 | **61.94** | **94.95** | **98.79** |

Table 5: Per-hierarchy comparison of mIoU on the `test` set of PASCAL-Person-Part (Xia et al., 2017). The backbone of all models is ResNet-101 (He et al., 2016).

| Method | Year | $mIoU^1$ | $mIoU^2$ | $mIoU^3$ |
|---|---|---|---|---|
| DeepLabV3+ (Chen et al., 2018) | ECCV18 | 67.84 | 84.01 | 94.55 |
| SPGNet (Cheng et al., 2019) | ICCV19 | 68.36 | — | — |
| PGN (Gong et al., 2019) | CVPR19 | 68.40 | — | — |
| CNIF (Wang et al., 2019) | ICCV19 | 70.76 | 84.80 | 95.18 |
| SemaTree (Ji et al., 2020) | ECCV20 | 71.59 | 85.44 | 95.98 |
| HHP (Wang et al., 2020b) | CVPR20 | 73.12 | 86.13 | 96.86 |
| BGNet (Zhang et al., 2020a) | ECCV20 | 74.42 | — | — |
| PCNet (Zhang et al., 2020b) | CVPR20 | 74.59 | — | — |
| HSSN (Li et al., 2022) | CVPR22 | 75.44 | 88.20 | 97.69 |
| HvsA | — | **76.37** | **88.94** | **97.88** |

$\mathcal{L}_{HM}$, our model yields substantial improvements, *e.g.*, $+10\%$ on Cityscapes. This suggests that our hierarchy-aware semantic learning is effective and essential for capturing the structure of semantic hierarchy. The results also imply that the two components well complement with each other.

**Geometric Space.** Next, we validate the choice of hyperbolic space by comparing with a variant of our model built in Euclidean space. The results are shown in Table 7. We observe that the model with hyperbolic space is consistently better than the Euclidean baseline. It appears that the performance gap is larger in image classification than semantic segmentation. We conjecture that the label taxonomy in image classification is more complex (*e.g.*, ImCLEF07A includes 96 classes, while Cityscapes only has 19), in which cases hyperbolic geometry becomes much more important in order to interpret the hierarchical structure.

**Hyperbolic Metric Learning.** Table 8 quantifies the effect of the coefficient $\alpha$ in $\mathcal{L}_{HM}$. The performance steadily improves as $\alpha$ increases, reaching the best performance when $\alpha = 0.2$. Additionally, as shown in Table 9, we compared the impact of different margins on the results. When $\gamma = 1$, the best performance is achieved, for higher value of $\gamma$, the results drop again. This is related to the exponential growth of the boundary expansion in hyperbolic space, which can accommodate embeddings for more categories. However, when the value of $\gamma$ is too large, it becomes hard to properly allocate the embedding to a space position that complies with the taxonomy, leading to poor convergence of the model.

**KL Divergence.** Table 10 illustrates how our model's performance varies with the coefficient $\beta$ of $D_{KL}$. Obviously, as $\beta$ becomes larger, the performance gradually improves and setting $\beta = 4.0$ provides a best result. When $\beta$ is relatively small, it leads to insufficient correlation between the posterior and prior. Conversely, the learnable prior distribution will impose wrong guidance on the posterior in the first few iterations, which leads to performance degradation.

Table 6: Key component analysis on Cityscapes and PASCAL-Person-Part.

| $\mathcal{L}_{\text{HM}}$ | $D_{\text{KL}}$ | Cityscapes | | PASCAL-Person-Part | | | ImCLEF07A | | | ImCLEF07D | | |
|---|---|---|---|---|---|---|---|---|---|---|---|---|
| | | mIoU[1] | mIoU[2] | mIoU[1] | mIoU[2] | mIoU[3] | MAP | CMAP | CV | MAP | CMAP | CV |
| | | 72.51 | 80.44 | 65.41 | 80.54 | 91.25 | 69.21 | 70.34 | 10.07 | 68.95 | 70.41 | 12.47 |
| | ✔ | 74.27 | 83.14 | 67.38 | 82.69 | 92.31 | 77.27 | 78.56 | 9.23 | 74.37 | 77.04 | 9.44 |
| ✔ | ✔ | **84.63** | **94.27** | **76.37** | **88.94** | **97.88** | **92.21** | **92.44** | **3.18** | **90.76** | **91.42** | **5.71** |

Table 7: Study of geometric space of latent space on Cityscapes (Cordts et al., 2016), PASCAL-Person-Part (Xia et al., 2017), ImCLEF07A and ImCLEF07D (Dimitrovski et al., 2011).

| Geometric space | Cityscapes | | PASCAL-Person-Part | | | ImCLEF07A | | | ImCLEF07D | | |
|---|---|---|---|---|---|---|---|---|---|---|---|
| | mIoU[1] | mIoU[2] | mIoU[1] | mIoU[2] | mIoU[3] | MAP | CMAP | CV | MAP | CMAP | CV |
| Euclidean | 83.04 | 92.41 | 74.17 | 87.11 | 96.42 | 88.47 | 90.14 | 5.47 | 87.61 | 88.47 | 6.98 |
| Hyperbolic | **84.63** | **94.27** | **76.37** | **88.94** | **97.88** | **92.21** | **92.44** | **3.18** | **90.76** | **91.42** | **5.71** |

Table 8: Analysis of $\alpha$ for $\mathcal{L}_{\text{HM}}$ on Cityscapes (Cordts et al., 2016), PASCAL-Person-Part (Xia et al., 2017), ImCLEF07A and ImCLEF07D (Dimitrovski et al., 2011).

| $\alpha$ | Cityscapes | | PASCAL-Person-Part | | | ImCLEF07A | | | ImCLEF07D | | |
|---|---|---|---|---|---|---|---|---|---|---|---|
| | mIoU[1] | mIoU[2] | mIoU[1] | mIoU[2] | mIoU[3] | MAP | CMAP | CV | MAP | CMAP | CV |
| 1.0 | 83.41 | 91.07 | 74.84 | 87.86 | 97.14 | 91.77 | 92.08 | 4.11 | 89.31 | 89.94 | 6.55 |
| 0.5 | 83.77 | 93.15 | 75.73 | 88.43 | 97.64 | **92.21** | 92.44 | **3.18** | 90.16 | 90.82 | 6.13 |
| 0.2 | **84.63** | **94.27** | **76.37** | **88.94** | **97.88** | 91.98 | **92.71** | 3.82 | **90.76** | **91.42** | **5.71** |
| 0.1 | 83.23 | 93.11 | 75.65 | 88.24 | 97.31 | 90.84 | 91.47 | 3.77 | 89.47 | 90.98 | 5.97 |

Table 9: Analysis of the margin $\gamma$ (Eq. 9) on Cityscapes (Cordts et al., 2016), PASCAL-Person-Part (Xia et al., 2017), ImCLEF07A and ImCLEF07D (Dimitrovski et al., 2011).

| $\gamma$ | Cityscapes | | PASCAL-Person-Part | | | ImCLEF07A | | | ImCLEF07D | | |
|---|---|---|---|---|---|---|---|---|---|---|---|
| | mIoU[1] | mIoU[2] | mIoU[1] | mIoU[2] | mIoU[3] | MAP | CMAP | CV | MAP | CMAP | CV |
| 2.0 | 83.94 | 94.03 | 76.12 | 88.14 | 97.76 | 90.48 | 90.76 | 4.44 | 90.23 | 90.94 | 5.97 |
| 1.0 | **84.63** | **94.27** | **76.37** | **88.94** | **97.88** | **92.21** | **92.44** | **3.18** | **90.76** | **91.42** | **5.71** |
| 0.5 | 81.06 | 90.34 | 74.01 | 85.97 | 94.43 | 90.71 | 91.05 | 4.59 | 88.37 | 90.11 | 6.37 |

Table 10: Analysis of $\beta$ for $D_{\text{KL}}$ on Cityscapes (Cordts et al., 2016), PASCAL-Person-Part (Xia et al., 2017), ImCLEF07A and ImCLEF07D (Dimitrovski et al., 2011).

| $\beta$ | Cityscapes | | PASCAL-Person-Part | | | ImCLEF07A | | | ImCLEF07D | | |
|---|---|---|---|---|---|---|---|---|---|---|---|
| | mIoU[1] | mIoU[2] | mIoU[1] | mIoU[2] | mIoU[3] | MAP | CMAP | CV | MAP | CMAP | CV |
| 5.0 | 83.07 | 92.15 | 75.67 | 86.67 | 96.11 | 90.35 | 91.07 | 3.76 | 89.32 | 90.26 | 6.14 |
| 4.0 | **84.63** | **94.27** | **76.37** | **88.94** | **97.88** | **92.21** | **92.44** | **3.18** | **90.76** | **91.42** | **5.71** |
| 3.0 | 83.76 | 93.24 | 74.36 | 85.72 | 96.37 | 91.07 | 91.76 | 3.81 | 89.96 | 90.55 | 6.22 |
| 2.0 | 83.13 | 92.47 | 73.44 | 85.26 | 95.93 | 88.14 | 89.07 | 4.12 | 86.14 | 87.04 | 6.71 |

## 6 CONCLUSION

This work proposes a solution for structural visual recognition from learning the shared representation of images and semantic concepts in hyperbolic space, yielding a new algorithm HVSA. HVSA leverages Gaussian mixture VAE to establish a representation space shared by two modes. Then, the hierarchical feature space is generated by hyperbolic metric learning. Through analysis, we confirm that HVSA's shared hyperbolic representation space helps to capture potential hierarchies. Its significant performance improvements and generalization capability are confirmed in a series of image-level and pixel-level benchmark tests.

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

This appendix provides theoretical proofs, additional results and experimental details for our paper – HYPERBOLIC VISUAL-SEMANTIC ALIGNMENT FOR STRUCTURAL VISUAL RECOGNITION.

# A EXPLANATION OF METHODOLOGICAL DETAILS

## A.1 ENTAILMENT CONES

Due to the significant volume occupied by each concept in the embedding space, attempting to achieve embeddings using a distance function leads to a severe problem of orthogonal overlap in the embedding space. To address this issue, (Ganea et al., 2018) introduced the concept of Euclidean cones which generalizes embeddings by substituting translated orthants with more flexible convex cones. Recently, researchers have explored more general and flexible methods that do not necessarily rely on Euclidean space and utilize cones in hyperbolic spaces to achieve the representation of hierarchical semantics.

**Euclidean Cones.** For each vector $x$ in $\mathbb{R}^N$, the aperture of the cone is based solely on the Euclidean norm of the vector, $||x||$, (Ganea et al., 2018) and is given by $\psi(x) = \arcsin(K/||x||)$ where K is a hyper-parameter. The cones can have a maximum aperture of $\pi/2$. To ensure continuity and transitivity, the aperture should be a smooth, non-increasing function. To satisfy properties mentioned in (Ganea et al., 2018), the domain of the aperture function has to be restricted to $(\varepsilon, 1]$ for some $\varepsilon$. $\varepsilon = f(K)$. Eq.13 computes the minimum angle between the axis of the cone at $x$ and the vector $y$. $E(x, y) = \max(0, \ \Xi(x, y) - \psi(x))$ measures the cone-violation which is the minimum angle required to rotate the axis of the cone at $x$ to bring $y$ into the cone.

$$\Xi(x, y) = \arccos \left( \frac{||y||^2 - ||x||^2 - ||x - y||^2}{2 \, ||x|| \, ||x - y||} \right) \tag{13}$$

**Hyperbolic Cones.** The Poincaré ball is defined by the manifold $\mathbb{D}^N = \{x \in \mathbb{R}^N : ||x|| < 1\}$. The distance between two points $x, y \in \mathbb{D}^N$ and the norm are

$$d_{\mathbb{D}}(x, y) = \text{arccosh}(1 + 2(||x - y||^2)/((1 - ||x||^2)(1 - ||y||^2)))$$

and $||x||_{\mathbb{D}} = d_{\mathbb{D}}(0, x) = 2 \, \text{arctanh}(||x||)$ where we use $||.||$ for Euclidean norm, $\langle ., . \rangle$ for dot-product and $\hat{x} = x/||x||$ for a unit vector. The angle between two tangent vectors $u, v \in T_x \mathbb{D}^n$ is given by $\cos(\angle(u, v)) = \langle u, v \rangle/(||u|| \, ||v||)$. The aperture of the cone is $\psi(x) = \arcsin(K(1 - ||x||^2)/||x||)$. $\Xi(x, y)$ computes the minimum angle between the axis of the cone at $x$ and the vector $y$.

$$\Xi(x, y) = \arccos\left( \frac{\langle x, y \rangle (1 + ||x||^2) - ||x||^2(1 + ||y||^2)}{\omega \sqrt{1 + ||x||^2 ||y||^2 - 2\langle x, y \rangle}} \right) \tag{14}$$

$E(x, y) = \max(0, \ \Xi(x, y) - \psi(x))$ measures the cone-violation which is the minimum angle required to rotate the axis of the cone at $x$ to bring $y$ into the cone. $\omega = ||x|| \, ||x - y||$

# B EXPERIMENTAL DETAILS

## B.1 DATASETS

We conduct extensive experiments on six datasets:

- **ImCLEF07A and ImCLEF07D** (Dimitrovski et al., 2011) represent medical X-ray images annotated with parts of the human anatomy and orientations of body parts. These two datasets consist of 7,000 training images, 3,000 validation images, and 1,006 test images each. The IMCLEF07A dataset encompasses a total of 96 different labels, while the IMCLEF07D dataset includes 46 labels, making it suitable for more challenging multi-label classification tasks.

- **TieredImageNet-H** (Bertinetto et al., 2020) is an extension of the TieredImageNet, designed to evaluate the performance of models in handling multi-label classification tasks with hierarchies of 13, covering 608 classes. The label taxonomy is illustrated in Fig. 2. This dataset comprises over 600,000 images, with 450,000 images in training set, 30,000 images in validation set, and 120,000 images in test set.

- **Cityscapes** (Cordts et al., 2016) is an urban scene parsing dataset with 5k finely annotated images, which contains 2,975/500/1,524 in train/val/test splits, respectively. The segmentation performance

is evaluated over 19 fine-grained concepts and 6 super-classes. We illustrate its label taxonomy in Fig. 3 (left).

- **LIP** (Liang et al., 2018a) is a large-scale human body parsing dataset including 50,462 single-person images with intimate pixel-wise annotations of 19 part categories. Holding by (Wang et al., 2019; 2020b; Li et al., 2022), the 19 subclasses are further reduced to two classes (`upper-body` and `lower-body`) and eventually to the `human-body` (see Fig. 3 (middle)). The images are divided into 30,462/10,000/10,000 images for train/val/test.

- **PASCAL-Person-Part** (Xia et al., 2017) contain 1,716 and 1,817 images in train and test split, respectively. It provides careful pixel-wise annotations for six body parts and has a similar hierarchy with LIP. The label taxonomy is shown in Fig. 3 (right).

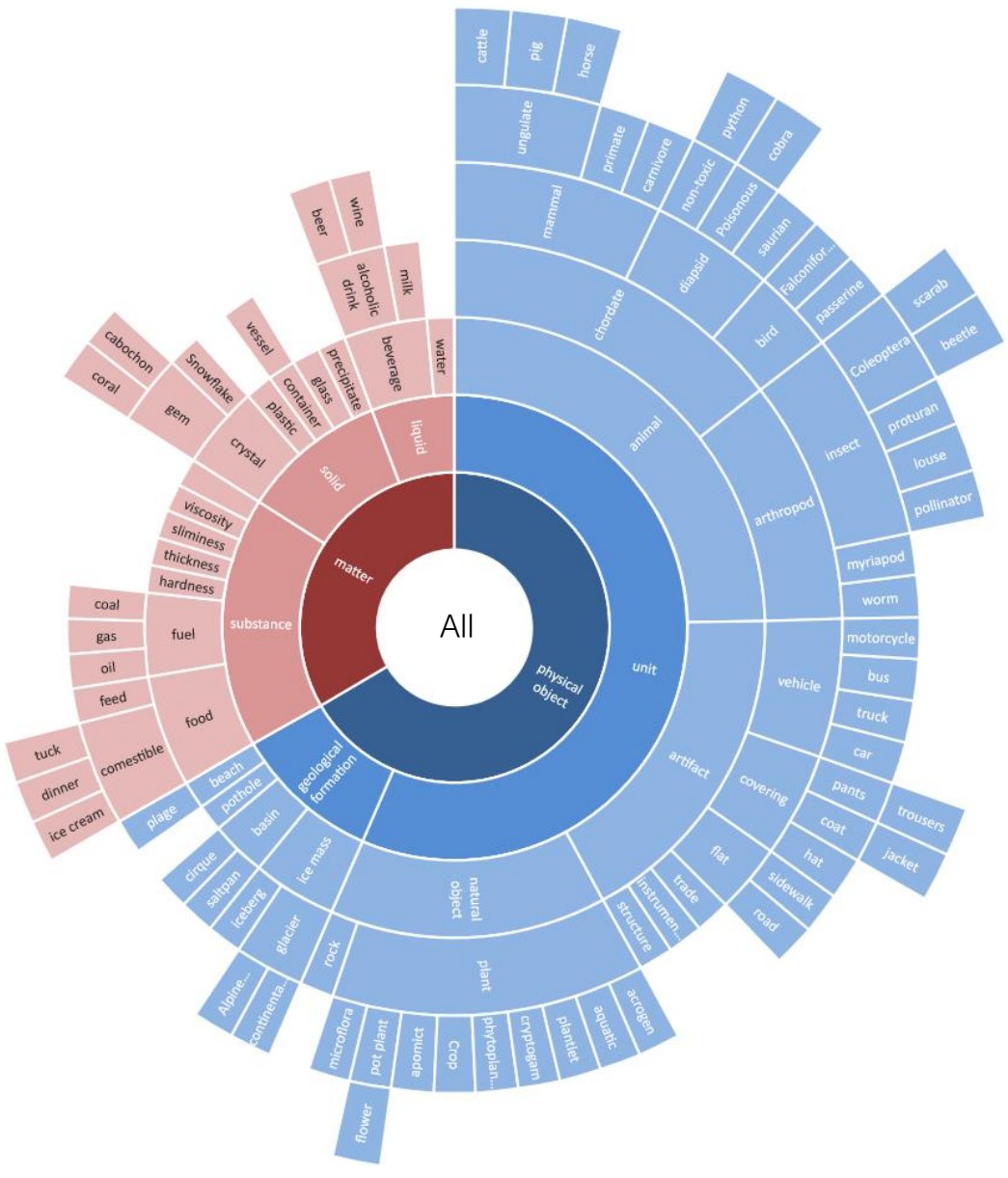

Figure 2: **Illustration of label taxonomy** in TieredImageNet-H (Bertinetto et al., 2020).

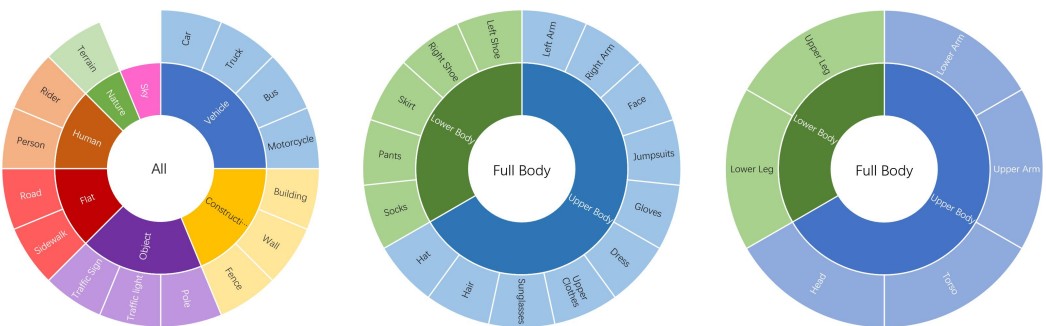

Figure 3: **Illustration of label taxonomy** in Cityscapes (left) (Cordts et al., 2016) , LIP (middle) (Liang et al., 2018a) and PASCAL-Person-Part (right) (Xia et al., 2017).

Table 11: Analysis of training/inference cost on Cityscapes (Cordts et al., 2016) val. (See §B.3)

| Method | mIoU$^1$ | mIoU$^2$ | # Param (M) | FLOPs | FPS |
|---|---|---|---|---|---|
| DeepLabV3+ (Chen et al., 2018) | 82.08 | 92.16 | 62.7 | 83.40G | 8.34 |
| HSSN (Li et al., 2022) | 83.02 | 93.31 | 64.3 | 87.39G | 6.38 |
| HvsA | 84.31 | 93.97 | 68.1 | 89.11G | 6.11 |

## B.2 METRICS

For image classification, we use top-1 error to fairly evaluate performance on tieredImageNet-H. And we employ the Mean Average Precision (MAP) metric to ensure a fair comparison. Additionally, we provide the evaluation results of Constraint Violation (CV) (Patel et al., 2022) and Mean Average Precision post Coherence correction (CMAP) (Patel et al., 2022) on ImCLEF07A and ImCLEF07D. Constraint violation is a punitive metric that employs latent label taxonomy to assess label prediction consistency. Lower CV values indicate higher predictive classification consistency. For Mean Average Precision post Coherence correction, given a complete label taxonomy, coherence can be imposed post-hoc by applying a modification. The CMAP value close to MAP value suggests that the model better captures the latent label hierarchy in the label space. Formally, CV and CMAP are defined as:

- **Constraint Violation.** Constraint Violatio serves as a penalizing metric, aiming to quantify the degree to which the label scores generated by the model deviate from the partial ordering of the latent label taxonomy, irrespective of the true labels associated with the instances.

$$CV = \frac{1}{|D||\mathcal{T}|} \sum_{k=1}^{|D|} \sum_{(v_i, v_j) \in \mathcal{T}} \mathbb{1}(s_i^{(k)} - s_j^{(k)} < 0) \qquad (15)$$

where $|D|$ represents the quantity of samples, and $s$ denotes the confidence score.

- **Mean Average Precision post Coherence correction.** In the context of a given complete label taxonomy $G$, post hoc consistency can be enforced by applying a modification function $\delta : \mathbb{R}^L \to \mathbb{R}^L$ to the label scores generated by the model. This modification ensures that for all $(l_i, l_j) \in \mathcal{T}$, it holds that $\delta(s_i) - \delta(s_j) < 0$. The specific strategy of the modification is to adjust the score $s_i$ for each label $l_i$ to be either the maximum score of any of its descendants $\delta_G^M$ in the taxonomy $G$ or the minimum score of its ancestors $\delta_G^m$ in the system. Specifically, for scores generated by the model, the definitions of these two modification functions are as follows:

$$\delta_G^m(s)_i = \min_{l_j \in \text{Anc}_G(l_i) \cup \{l_i\}} s_j, \qquad \delta_G^M(s)_i = \max_{l_j \in \text{Des}_G(l_i) \cup \{l_i\}} s_j \qquad (16)$$

where $\text{Anc}_G(l)$ and $\text{Des}_G(l)$ denote the sets of ancestors and descendants of label $l$ in the graph $G$, respectively

As conventions, we adopt the mean intersection-over-union (mIoU) for semantic segmentation. And following (Li et al., 2022), we also report mIoU$^l$, the score for each hierarchy level $l$. For hierarchy-agnostic methods, the scores of each non-leaf layer calculated by combining the segmentation predictions of its subclasses.

### B.3 TRAINING AND INFERENCE COMPUTATION EFFICIENCY

Table 11 evaluates the training and inference efficiency against two methods HSSN (Li et al., 2022) and DeepLabV3+ (Chen et al., 2018) on Cityscapes val, in terms of FLOPs, FPS, and the number of parameters. As seen, our model basically exhibits a similar level of training and inference efficiency as the other two methods. Though it has more parameters, a larger FLOPs, and runs slower (smaller FPS), the gap is minor. Considering the superior performance and generalization capability of our model, the additional cost is acceptable.

### B.4 VISUALIZATION

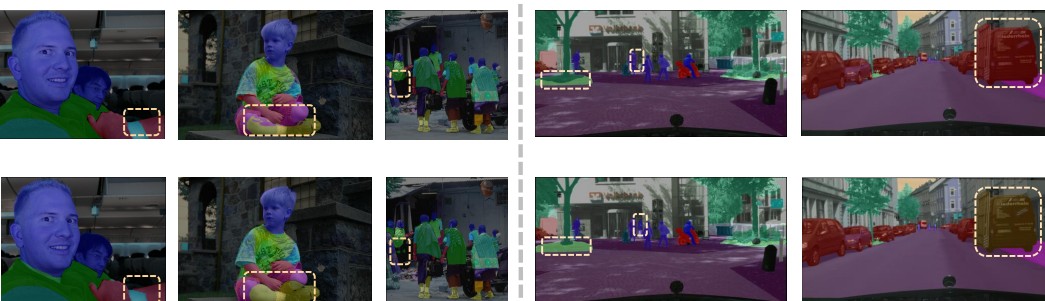

Figure 4: **Visual results** on PASCAL-Person-Part (Xia et al., 2017)(Left) and Cityscapes (Cordts et al., 2016). Top: HSSN (Li et al., 2022), Bottom: HVSA.

In Fig.4, we provide some visualization comparison between HSSN with our method on two datasets. HVSA achieves good performance in the scene of occlusion and tiny targets, showing the robustness. As shown in the second and the last column, major mistakes such as misclassifies happen in the results of HSSN. However, HVSA handles the severe mistakes, which indicates that compared to imposing hierarchical constraints in Euclidean space, our method is more effective in learning hierarchical knowledge among labels. Additional visualization results are provided in Fig 5 and Fig 6. We present the prediction results of the baseline method HSSN (left) and HVSA (right). It can be seen that, compared to the baseline model, the results of HVSA consistently align better with the hierarchical visual world.

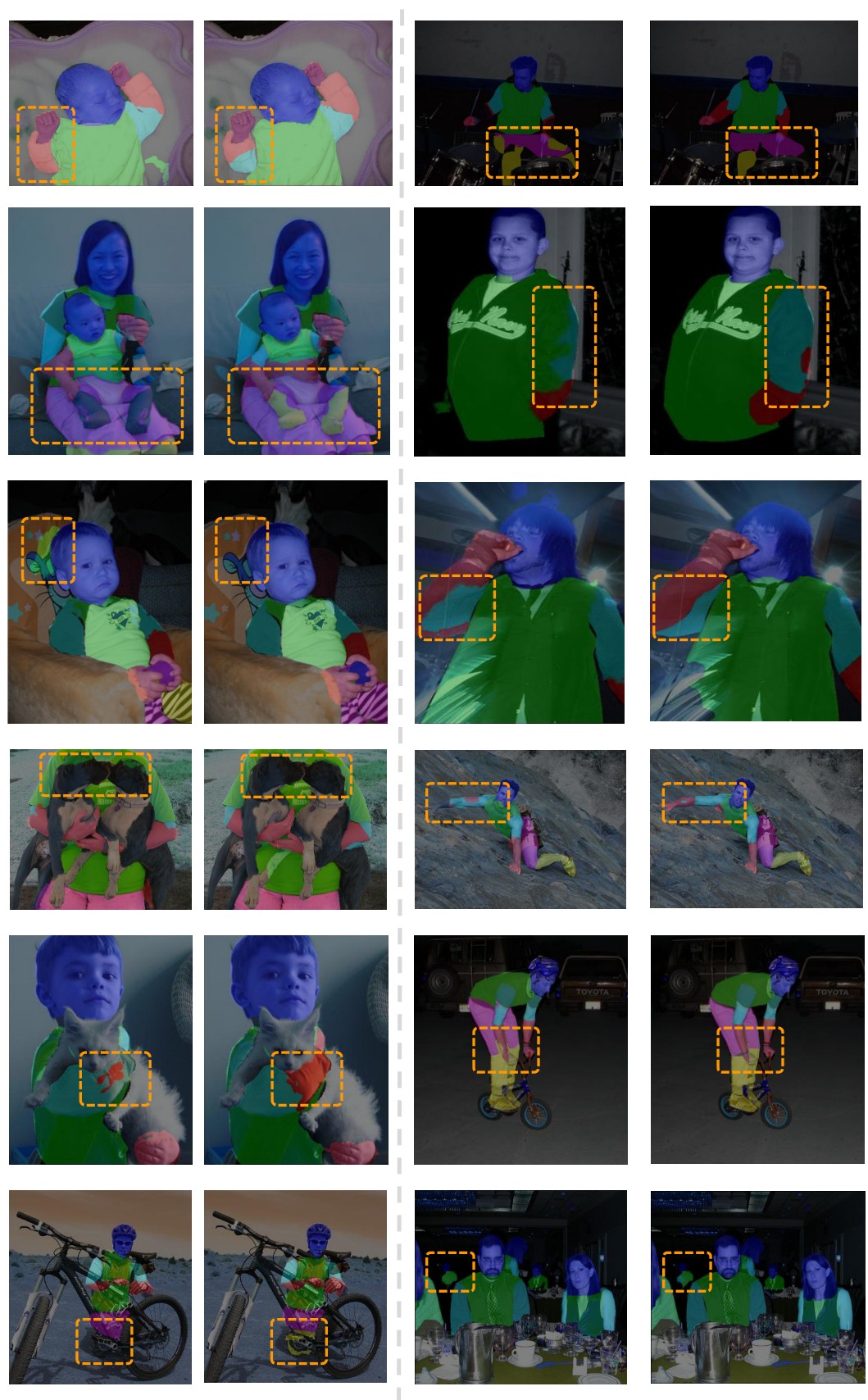

Figure 5: **Visual results** on PASCAL-Person-Part (Xia et al., 2017).

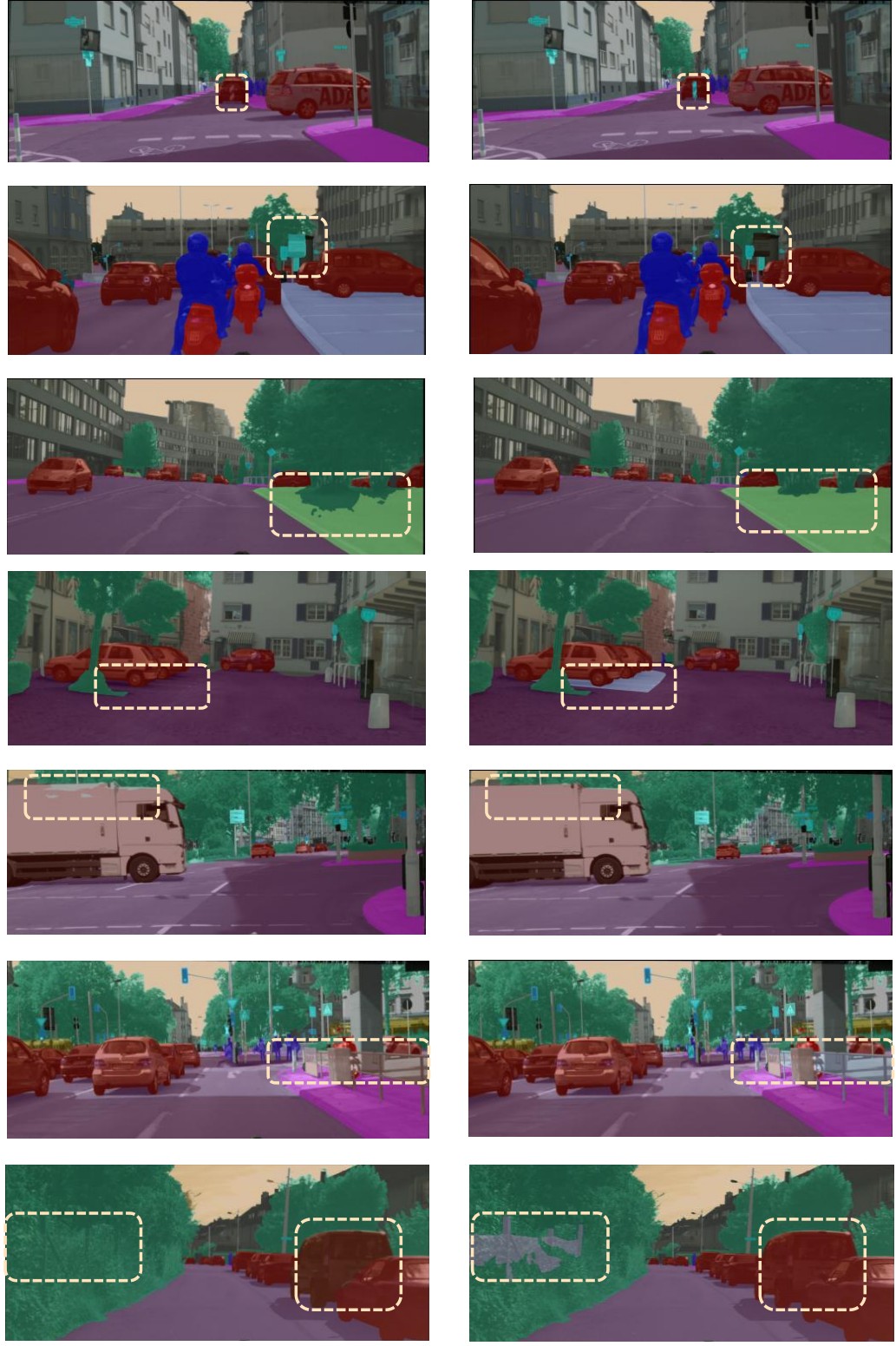

Figure 6: **Visual results** on Cityscapes (Cordts et al., 2016).

