# OpenReview forum: "Hyperbolic Visual-Semantic Alignment for Structural Visual Recognition"
_ICLR.cc/2024/Conference — ICLR 2024 Conference Withdrawn Submission_

### Official Review · Reviewer_SCte · 2023-10-31

**Soundness:** 3 good
**Presentation:** 3 good
**Contribution:** 3 good
**Rating:** 6
**Confidence:** 3

**Summary:**

This paper introduces a method named HvsA for structural visual recognition. HvsA consists of two stages. In “hierarchy-agnostic visual-semantic alignment”, it first encodes images and semantic labels in the hyperbolic space. Then it utilizes a Gaussian mixture VAE to align visual embedding with label embedding to yield a shared representation space. While in “hierarchy-aware semantic learning”, it leverages triplet metric learning over label space to push the label embeddings away for those without hierarchical relationships while pull close those with similar semantics. Experiments are conducted over six datasets, showing the effectiveness of the method.

**Strengths:**

1. The paper is well-written and easy to follow.
2. The concept of encoding hierarchical label embeddings and aligning visual embeddings is reasonable. Furthermore, the introduction of a triplet loss designed to operate within the label space, thereby facilitating the alignment of visual embeddings in a more hierarchical manner, is sound.
3. The experimental results are competitive.

**Weaknesses:**

1. The definition of q_θ in Eq 7 and 8 are missing.
2. The training and inference computation efficiency is not presented.
3. Minor issue:
  3.1. Typo: Section 3.2, Task Setting: “An undirected edges (vi, vj) ∈ E indicates that the class i is a superclass of label i” It should be “label j” instead of “label i”.
  3.2. Typo: Section 3.2.1 Hyperbolic Gaussian Mixture VAE: “we seek to align visual embedding with visual embedding to yield a shared representation space.” Is it “align visual embedding with LABEL embedding”?

**Questions:**

Are the authors planning to release the source code for this work?

---

> ### Author Response · Authors · 2023-11-21
> **Point-to-Point Response to Reviewer SCte**
>
> We thank the reviewer for the feedback and suggestions, and appreciate the positive comments. We address your comments below, and start with the following question:
>
> > 1. The definition of $q_{\phi}$ in Eq 7 and 8 are missing.
>
> Sorry about this. Here we follow the conventional notation in VAE [ref1]:  $q_{\phi}(z|x)$ represents the variational posterior distribution that  is an
> approximation to the intractable true posterior $p_\theta(z|x)$. It typically refers to a probabilistic encoder, which produces a distribution (in our paper is a Gaussian) of latent representation $z$ given a feature $x$. Relevant parts have been carefully revised to reflect this.
>
> [ref1] Auto-Encoding Variational Bayes. ICLR 2014
>
> > 2. The training and inference computation efficiency is not presented.
>
> To address your concern, we provide comparisons of training/inference cost in below table on Cityscapes val. The table has been incorporated into Appendix Table 11.
>
> |Method |  mIoU$^1$, mIoU$^2$  | #Param (M) | FLOPs (G) | FPS |
> |:----------|:------------:|:------------:|:---------:|---------:|
> | DeepLabV3+  | 82.08, 92.16 | 62.7 | 83.40 | 8.34 |
> | HSSN  | 83.02, 93.31 | 64.3 | 87.39 | 6.38 |
> | HvsA  | 84.31, 93.97 | 68.1 | 89.11 | 6.11 |
>
> As seen, our model basically exhibits a similar level of training and inference efficiency as the other two methods. Though it has more parameters, a larger FLOPs, and runs slower (smaller FPS), the gap is minor. Considering the superior performance and generalization capability of our model, the additional cost is acceptable.
>
>
> > 3. Minor issue:
> > 3.1 Typo: Section 3.2, Task Setting: “An undirected edge $(v_i, v_j) \in E$ indicates that the class $i$ is a superclass of label $i$” It should be “label $j$” instead of “label $i$”. 3.2 Typo: Section 3.2.1 Hyperbolic Gaussian Mixture VAE: “we seek to align visual embedding with visual embedding to yield a shared representation space.” Is it “align visual embedding with LABEL embedding”?
>
> Thank you so much for your careful review! They are fixed and a thorough proofreading is made.
>
> “An undirected edge $(v_i, v_j) \in E$ indicates that the class $i$ is a superclass of label $i$” $\rightarrow$ “An undirected edge $(v_i, v_j) \in E$ indicates that the class $j$ is a superclass of _label $i$_”
>
> “we seek to align visual embedding with visual embedding to yield a shared representation space.” $\rightarrow$ “we seek to align visual embedding with _label_ embedding to yield a shared representation space.”
>
> > Are the authors planning to release the source code for this work?
>
> Definitely! We will release the code and model weights to the public.

---

### Official Review · Reviewer_Lirp · 2023-10-31

**Soundness:** 2 fair
**Presentation:** 3 good
**Contribution:** 2 fair
**Rating:** 3
**Confidence:** 4

**Summary:**

The authors of the paper introduced a method for visual-semantic alignment in hyperbolic space. They presented a hierarchy-agnostic approach to visual-semantic alignment, complemented by hierarchy-aware semantic learning. To demonstrate the effectiveness of their proposed model, experiments were carried out on classification and segmentation tasks across six datasets.

**Strengths:**

1. The visual-semantic alignment from a probabilistic perspective in hyperbolic space is novel.
2. The hierarchy-agnostic idea is innovative.
3. Extensive experiments were conducted, covering multiple tasks.

**Weaknesses:**

1. The logic flow is broken, one can hardly see the connection between 3.1, 3.2. It will be very difficult (even for a hyperbolic learning researcher) to reproduce the method by reading the ``Method'' section.

2. Many important descriptions are missing or even incorrect, e. g. ,

>  " ... the mean of the wrapped normal mixture distribution is calculated by Möbius addition. " is not logically correct.

>  " ... resulting in $y_{l,i} = \mathbf{\mu}_l + \mathbf{\Sigma}_l$ ...", you cannot add a mean vector to a covariance matrix.

>  The reconstruction loss is neither explained nor linked to a reference

3. The tables basically show "we have better numbers", but the analysis lacks in-depth understanding of why each part works.

**Questions:**

Please see the weaknesses. besides, I have one more question regarding the motivation

> What is the aim of hierarchy-agnostic alignment when we already have the hierarchy information? Why is the ``agnostic'' part important?

---

> ### Author Response · Authors · 2023-11-21
> **Point-to-Point Response to Reviewer Lirp**
>
> We appreciate the reviewer for pointing out certain issues of clarity in the technical part which we have revised in the latest revision and provide clarifications below.
>
> To begin with, we address the following question that is relevant to our motivation:
>
> > What is the aim of hierarchy-agnostic alignment when we already have the hierarchy information? Why is the ``agnostic'' part important?
>
> Sorry for the confusion. Please bear with our clarifications below.   Our main idea is to learn a shared, hierarchical, multimodal feature space. Our model achieves this based on two major insights:  (1) each visual entity (e.g., image, pixel) can be represented through a composition of multiple semantic labels, and (2) these labels are correlated in nature, collectively forming a tree-shaped hierarchy. Based on these insights, we develop two components: hierarchy-agnostic visual-semantic alignment addresses (1), and hierarchy-aware semantic learning addresses (2).
>
> **Difference and relation between the two components:** the hierarchy-agnostic component __handles multimodal alignment, but ignores hierarchical information__; the hierarchy-aware component __only interprets the hierarchical structure of semantic concepts (visual information is not taken into account)__. These two objectives operate on different granularity and synergistically contribute visual-semantic structural alignment. Notably, direct aligning the hierarchical structures between two modalities is very challenging, and our decoupled design alleviates this and helps model converge.
>
> Next, we address your comments on paper writing:
>
> > The logic flow is broken, one can hardly see the connection between 3.1, 3.2. It will be very difficult (even for a hyperbolic learning researcher) to reproduce the method by reading the ``Method'' section.
>
> Our apologies. We agree that the writing of mentioned parts can be improved upon rereading.  We have worked to improve our writing.
>
> - _connection between 3.1 and 3.2_: Sec. 3.1 is only to present the basic formulations for hyperbolic learning. It helps readers to understand basic concepts, and the formulations are also reused in Sec. 3.2 (_e.g.,_ Eq. 5, Eq. 6, Eq. 8). There are no inherent connections between them. To avoid confusion, we take an action to move Sec. 3.1 into a new Sec. 3 and the methodology part into a new Sec. 4. Thanks.
>
> - writing improvements in the methodology part:
>
>   (1) in Sec. 4.1, we have added missed details, including an explicit definition of covariance, i.e., $\Sigma_l=diag(\sigma_l^2)$, as well as the definition of the reconstruction loss.
>
>   (2) in Sec. 4.2, we re-written most parts, and make the section more logical. Concretely, we compile hierarchy-aware semantic learning into two major parts, one is hyperbolic metric learning, and the other is how to construct samples for metric learning.
>
> We believe that these improvements make the methodology more logical and easier to understand. Beyond this, we promise that we will release the code to help  reproducibility. Thanks.
>
> > a) " ... the mean of the wrapped normal mixture distribution is calculated by Möbius addition. " is not logically correct.
>
> We agree with the reviewer: this statement is not reasonable and is redundant; we removed it in the revised version.
>
> > b) " ... resulting in $y_{l,i} = \mu_l + \epsilon\Sigma_l \in \mathcal{Y}$...", you cannot add a mean vector to a covariance matrix.
>
> Thank you for your careful review. In the original writing, we accidentally omitted the explicit definition of $\Sigma_l$, which is a diagonal matrix: $\Sigma_l=diag(\sigma_l^2)$. Here $\mu_l\in\mathbb{R}^d$ and $\sigma_l\in\mathbb{R}^d$ are derived from the semantic encoder. With this respect, the formula of the reparameterization trick should be more strictly revised to $y_{l,i} = {\mu}_l + \epsilon\sigma_l$. The relevant parts have been carefully revised.
>
> > c) The reconstruction loss is neither explained nor linked to a reference
>
> The reconstruction loss is a standard negative log-likelihood. We provide its formal definition in Eq. 8 in the updated article. Thanks.
>
> > The tables basically show "we have better numbers", but the analysis lacks in-depth understanding of why each part works.
>
> Thanks for pointing this out. Following your comments, we have carefully improved the experimental parts to offer more in-depth discussions on the results. Please refer to Sec. 5.2 and Sec. 5.3 for details.

---

> > ### Comment · Reviewer_Lirp · 2023-11-27
> >
> > Thank you for the effort you have put into revising your manuscript. I would like to thank the authors for the changes they have made since the first review. I also read the other reviewers comments, and I would like to stay with my initial score.
> >
> > My main concern remains. The contribution of this paper seems primarily to be a combination of methods that already exist. While combining existing techniques can be valuable, it is important to show how this new combination creates something significantly different or better. In its current form, the paper does not clearly demonstrate this.
> >
> > Furthermore, there seems to be a lack of in-depth analysis of the real driving factors. Such an analysis can provide valuable insights and strengthen the significance of your work. As of now, this deeper understanding or explanation of the underlying factors is still insufficient.

---

> > > ### Author Response · Authors · 2023-11-28
> > > **Further discussions**
> > >
> > > Thanks for your response to our rebuttal.
> > >
> > > > My main concern remains. The contribution of this paper seems primarily to be a combination of methods that already exist. While combining existing techniques can be valuable, it is important to show how this new combination creates something significantly different or better. In its current form, the paper does not clearly demonstrate this.
> > >
> > > First of all, we are sorry but indeed somewhat confused about this newly raised issue. In the first-round review, the reviewer has mentioned in [Strengths] that the proposed major components are novel: ``The visual-semantic alignment from a probabilistic perspective in hyperbolic space is novel``, ``The hierarchy-agnostic idea is innovative``, but never mentioned that our method is a combination of others. It is hard for us to address the "main concern" if it is not written in the comments.
> > >
> > > **We further explain our idea to hopefully help understanding.** The main idea of the paper is to learn a shared, hierarchical, multimodal feature space. While directly learn such a feature space is challenging, we attempt to decouple the problem based on two insights: (1) each visual entity (e.g., image, pixel) can be represented through a composition of multiple semantic labels, and (2) these labels are correlated in nature, collectively forming a tree-shaped hierarchy. As a result, the hierarchy-agnostic component is introduced to solely handle multimodal alignment, ignoring hierarchical information, while the hierarchy-aware component only interprets the hierarchical structure of semantic concepts (visual information is not taken into account). These two objectives operate on different granularity and synergistically contribute visual-semantic structural alignment.
> > >
> > > We did not notice any previous works offering similar insights or working in a similar manner. For evaluation, we have provided extensive experiments including ablation study on four datasets. The performance improvements are notable and consistent. We believe that the efficacy is properly proved.
> > >
> > > > Furthermore, there seems to be a lack of in-depth analysis of the real driving factors. Such an analysis can provide valuable insights and strengthen the significance of your work. As of now, this deeper understanding or explanation of the underlying factors is still insufficient.
> > >
> > > Thanks for your comments on this. Actually in the **first two paragraphs of the Introduction**, we have clearly presented the real driving factors that motivate our work. There are two major factors: (1) hierarchical structures are nature of many practical problems (**the first paragraph**), and (2) humans can easily understand such hierarchies and thus it is essential to develop AI algorithms to mimic such behaviors (**the second paragraph**). In addition, the importance and potential of the task are also stated as "Hierarchical representations can not only properly handle such open-world cases, but also show the potential to improve interpretability (Nauta et al., 2021) and enable better exploratory data analysis of large datasets (Deng et al., 2009)."  (**the second paragraph**).
> > >
> > > We will follow the reviewer's suggestion to expand these paragraphs so as to provide more in-depth discussions about these driving factors. Thanks.

---

> > > > ### Comment · Reviewer_Lirp · 2023-11-28
> > > >
> > > > Thanks for the elaboration.
> > > >
> > > > I understand the novelty of the proposed concept, but the concept itself is not sufficient to prove its contributions. especially when the proposed models are **a combination of existing models**, eq.(4) through eq.(11) are from existing literature, while eq.(9) to eq.(11) are almost the same as in (Ganea et al., 2018), while it is cited there.
> > > >
> > > > Moreover, I have questions regarding the authors insights in their reply:
> > > >
> > > > - The two insights the authors mentioned in the reply were likely already examined by previous research in the references:
> > > >
> > > > ``(1) Each visual entity (e.g., image, pixel) can be represented through a composition of multiple semantic labels``. seems to be examined by (Liu et al., 2020), Maybe can the authors be more specific on what does ``a composition of multiple semantic labels`` mean ?
> > > >
> > > > ``(2) These labels are correlated in nature, collectively forming a tree-shaped hierarchy.`` is examined by (Ganea et al., 2018) and (Liu et al., 2020).
> > > >
> > > > - The in-depth analysis should be supported by designed experiments instead of narrations. Showing **WHY the proposed model works** brings way more contribution than ``we have better numbers``.
> > > >
> > > > For example, the proposed component,  ``Probabilistic Label Embedding`` part is even not supported by ``we have better numbers``. The authors mention that the proposed ``Probabilistic Label Embedding'' is fundamentally different to those non-probabilistic embedding models, e.g., (Liu et al., 2020) and (Long et al., 2020). However, there isn't a comparison against those baselines. leaving the proposed model enigmatic to the audiences.

---

> > > > > ### Author Response · Authors · 2023-11-29
> > > > > **reply**
> > > > >
> > > > > > I understand the novelty of the proposed concept, but the concept itself is not sufficient to prove its contributions. especially when the proposed models are a combination of existing models, eq.(4) through eq.(11) are from existing literature, while eq.(9) to eq.(11) are almost the same as in (Ganea et al., 2018), while it is cited there.
> > > > >
> > > > > First of all, we highlight that NOVELTY does not mean we never use existing techniques.
> > > > >
> > > > > For "eq.(4) through eq.(11) are from existing literature", **we believe that the novelty of the idea has been confirmed by the reviewer. Though each single technique (VAE, hyperbolic learning, Gaussian mixture model) is not new, our approach integrating them yields a new solution, at least for the task of hierarchical recognition.**
> > > > >
> > > > > For "eq.(9) to eq.(11) are almost the same as in (Ganea et al., 2018)", true, we use a similar loss function as in (Ganea et al., 2018), but **our approach differs from it in two major aspects:** a) in our approach, label embeddings are modeled in a probabilistic manner, while (Ganea et al., 2018) is a non-probabilistic model; b) how label embeddings are sampled: (Ganea et al., 2018) only samples linked concepts in the hierarchy as anchor-positive pairs, while our approach does not have such constraints.
> > > > >
> > > > > > Maybe can the authors be more specific on what does a composition of multiple semantic labels mean?
> > > > >
> > > > > Sorry for the confusion. Here we mean that in hierarchical recognition tasks, each visual entity (e.g., image, pixel) corresponds to multiple semantic labels. For example, an image with a bus might have three labels "vehicle, large-vehicle, bus" in Cityscapes taxonomy. This actually reveals the multi-label classification nature of the task.
> > > > >
> > > > > > "(1) Each visual entity (e.g., image, pixel) can be represented through a composition of multiple semantic labels." seems to be examined by (Liu et al., 2020).
> > > > >
> > > > > This is a misleading. The work (Liu et al., 2020) only studied hierarchical image embedding learning, and never explicitly addressed the composition issue discussed above. In contrast, our approach addresses this through a Gaussian mixture model, in which each label embedding is modeled as a Gaussian. This solution is completely different from the one in (Liu et al., 2020).
> > > > >
> > > > > > (2) These labels are correlated in nature, collectively forming a tree-shaped hierarchy. is examined by (Ganea et al., 2018) and (Liu et al., 2020).
> > > > >
> > > > > Yes. This is the nature of the hierarchical recognition task, which has been extensively studied in the literature beyond the two works the reviewer mentioned. What we claim in the previous reply is not rigorous. We intended to express that our approach is new in learning a shared, hierarchical, multimodal feature space by addressing the two insights separately.
> > > > >
> > > > > > The in-depth analysis should be supported by designed experiments instead of narrations. Showing WHY the proposed model works brings way more contribution than we have better numbers.
> > > > >
> > > > > > For example, the proposed component, Probabilistic Label Embedding part is even not supported by we have better numbers. The authors mention that the proposed ``Probabilistic Label Embedding'' is fundamentally different to those non-probabilistic embedding models, e.g., (Liu et al., 2020) and (Long et al., 2020). However, there isn't a comparison against those baselines. leaving the proposed model enigmatic to the audiences.
> > > > >
> > > > > Thanks for your suggestion.  In the first-round review, all reviewers have confirmed that our experiments are extensive. We have provided a detailed ablation study to provide in-depth understanding of the model.
> > > > >
> > > > > For your proposal to compare against (Liu et al., 2020) and (Long et al., 2020), these two works solve very different tasks as us, i.e.,  (Liu et al., 2020) focuses on zero-shot recognition, (Long et al., 2020) focuses on video action recognition. It is hard to directly apply them in our setting, but we will investigate how to adapt them to our problem in the next version. Thanks.

---

> ### Comment · Reviewer_Lirp · 2023-11-29
>
> - In response to ``For eq. (9) to eq. (11) are almost the same as in (Ganea et al., 2018), true, we use a similar loss function as in (Ganea et al., 2018), but our approach differs from it in two major aspects ...``
>
> In the submission, eq. (9) is in essence the same as eq. (32) in (Ganea et al.. 2018), eq. (10) is in essence the same as eq. (33) in (Ganea et al., 2018). The only notation difference is that
> 1. The authors use $(z_a, z_p) \in P$ for positive pairs and $(z_a, z_n)\in N$ for negative pairs
> 2. (Ganea et al., 2018) uses $(u, v) \in P$ for positive pairs and $(u', v') \in N$ for negative pairs.
>
> In the submission, eq. (11) is in **EXACTLY the same** as eq.(28) in (Ganea et al., 2018).
>
> However, in (Ganea et al., 2018), eq. (28) and eq. (33) is used for defining the cone, which is the **core contribution, such that the word "cone" is in the title of   (Ganea et al., 2018)**.
>
> In this submission, those cone definitions are neither related to "visual-semantic alignment from a probabilistic perspective" nor "hierarchy-agnostic perspective", which I found conceptually novel.
>
> - In response to ``The work (Liu et al., 2020) only studied hierarchical image embedding learning, and never explicitly addressed the composition issue discussed above, this solution is completely different from the one in (Liu et al., 2020).``
>
> In the authors' last comments, they defined the compositional label with an example: ``a bus might have three labels "vehicle, large-vehicle, bus" in Cityscapes taxonomy``, which is the ``grantparent_class`` to ``parent_class`` to ``children_class`` relationship on the hierarchy.  (Liu et al., 2020) did the same, but on the WordNet hierarchy, I didn't see the authors fundamental differences.
>
> - In response to ``all reviewers have confirmed that our experiments are extensive. We have provided a detailed ablation study to provide in-depth understanding of the model.``
>
> I would like to reiterate that ``"we have better numbers" is NOT in-depth analysis``. One of the authors' core contributions, ``Probabilistic Label Embedding``'s effectiveness against non-probabilistic embeddings, is NOT supported by any of those 10 Tables.

---

> > ### Author Response · Authors · 2023-11-30
> > **reply**
> >
> > We are grateful for the reviewer's engagement. Before addressing specific points, we'd like to note the importance of understanding our contributions from a holistic manner, rather than isolated examination of Sec. 4.1 and Sec. 4.2. These two parts respectively delve into multi-modality, hierarchy-agnostic (Sec. 4.1) and single-modality, hierarchy-aware (Sec. 4.2) learning, which complement to each other both intuitively and empirically. The holistic view makes it easier to differentiate our approach from prior studies within the field of hierarchical recognition.
> >
> > > In the submission, eq. (9) is in essence the same as eq. (32) in (Ganea et al.. 2018), eq. (10) is in essence the same as eq. (33) in (Ganea et al., 2018). The only notation difference is that ...
> > > In the submission, eq. (11) is in EXACTLY the same as eq.(28) in (Ganea et al., 2018).
> >
> > Thanks for your comments. As in the previous reply, we **indeed follow (Ganea et al.. 2018) for the loss function and have properly cited the paper in the article. However, they are not identical but differ in two aspects**: 1) label embeddings in our method is probabilistic, and 2) we sample anchor-positive samples randomly, rather than only linked categories in the hierarchy.
> >
> > We kindly refer the reviewer to the text description in Sec. 4.2, rather than only comparing the equations. Thanks.
> >
> > > In the authors' last comments, they defined the compositional label with an example: a bus might have three labels "vehicle, large-vehicle, bus" in Cityscapes taxonomy, which is the grantparent_class to parent_class to children_class relationship on the hierarchy. (Liu et al., 2020) did the same, but on the WordNet hierarchy, I didn't see the authors fundamental differences.
> >
> > Sorry for the confusion. Here the fundamental difference is that our approach in Sec. 4.1 is **_hierarchy-agnostic_, which means we ignore the hierarchical structure in the compositional labels, e.g., "vehicle, large-vehicle, bus" are treated as _disjoint_.** In contrast, (Liu et al., 2020), based on the WordNet hierarchy, is only aware of hierarchy learning.
> >
> > > I would like to reiterate that "we have better numbers" is NOT in-depth analysis. One of the authors' core contributions, Probabilistic Label Embedding's effectiveness against non-probabilistic embeddings, is NOT supported by any of those 10 Tables.
> >
> > Thanks for your comments. We have started working on this experiment, and will update to you ASAP after finished.

---

> ### Comment · Reviewer_Lirp · 2023-11-30
>
> Thanks for the quick reply.
>
> - In response to ``... we indeed follow (Ganea et al.. 2018) for the loss function and have properly cited the paper in the article. However, they are not identical but differ in two aspects: 1) label embeddings in our method is probabilistic, and 2) we sample anchor-positive samples randomly...``
>
> Both of the authors' statements do not hold:
>
> 1. For ``label embeddings in our method is probabilistic`` is defined by eq. (3) and eq. (4) in the authors' submission, NOT in eq. (9) - eq. (11) that we are discussing.
>
> 2. For ``we sample anchor-positive samples randomly, rather than only linked categories in the hierarchy.`` the difference is minor, and the effect of this difference is NOT supported by any of the 10 tables.
>
> To make it more specific, in (Ganea et al.. 2018), Ganea used very subtle controls over the sampling probability to control the ratio of positive and negative samples, I strongly doubt that generalizing their sampling choice to `` classes with higher semantic similarities (closer in the tree T ) to anchor are selected as positive samples`` would have a positive effect on the embedding quality.
>
> - In response to ``...our approach in Sec. 4.1 is hierarchy-agnostic, which means we ignore the hierarchical structure in the compositional labels, e.g., "vehicle, large-vehicle, bus" are treated as disjoint.``
>
> The word ``composition`` or ``compositional`` appears ZERO times in Section 4.1, while in Section 4.2, the authors claim ``§4.1 addresses the compositional properties of semantic concepts solely...``. If it is one of the major differences from the existing methods, it is NOT supported by any of the sections, and it is also NOT supported by any of the 10 tables.
>
> Also, the authors' comments CONTRADICTS with the last paragraph of the related work in their submission, where they wrote, ``... Second, compositionality: hierarchies often emerge from the composition of basic elements. This observation has driven prior work to apply hierarchical representations learned in hyperbolic space, such as ... segmentation (Weng et al., 2021) and action recognition (Long et al., 2020).``, where the references DO NOT treat the ``compositional labels`` as disjoint labels, instead they treat them as hierarchical labels.

---

> > ### Author Response · Authors · 2023-11-30
> > **reply**
> >
> > > For label embeddings in our method is probabilistic is defined by eq. (3) and eq. (4) in the authors' submission, NOT in eq. (9) - eq. (11) that we are discussing.
> >
> > True. Probabilistic label embedding is defined in Sec. 4.1, but that does not mean it only works for Sec. 4.1. It works throughout the article including Sec. 4.2, please see "Constructing Samples for Metric Learning" in Sec. 4.2. Thanks.
> >
> > > The word composition or compositional appears ZERO times in Section 4.1, while in Section 4.2, the authors claim §4.1 addresses the compositional properties of semantic concepts solely.... If it is one of the major differences from the existing methods, it is NOT supported by any of the sections, and it is also NOT supported by any of the 10 tables.
> >
> > Sorry for the confusion. In the Introduction section, we have clearly explained that we create "hierarchy-agnostic visual-semantic alignment", which corresponds to Sec. 4.1, to tackle the issue of label composition. This is also evident from the **mixture** formulation of multiple Gaussian distribution. We didn't perceive any issue without using the word "composition" in Sec. 4.1.
> >
> > > Also, the authors' comments CONTRADICTS with the last paragraph of the related work in their submission, where they wrote, ... Second, compositionality: hierarchies often emerge from the composition of basic elements. This observation has driven prior work to apply hierarchical representations learned in hyperbolic space, such as ... segmentation (Weng et al., 2021) and action recognition (Long et al., 2020)., where the references DO NOT treat the compositional labels as disjoint labels, instead they treat them as hierarchical labels.
> >
> > This is not a contradiction. But we found that the sentence "hierarchies often emerge from the composition of basic elements" can be revised to avoid confusion upon rereading. We will revise it to "hierarchies often emerge from a structured composition of basic elements".
> >
> > While it is straightforward to treat concepts as hierarchical labels as in (Weng et al., 2021), (Long et al., 2020), our solution just works distinctly, by addressing the "composition" and "hierarchy/structured" separately, in Sec. 4.1 and Sec. 4.2.

---

> ### Comment · Reviewer_Lirp · 2023-11-30
>
> Thanks for the reply.
>
> - It is completely fine to use words other than ``composition`` to describe the concept. However, as one of the major concepts that is different from existing work, ``compositional labels`` is neither explained, nor explored in Section 4.1. Sections 4.1 only has three parts ``Hyperbolic Probabilistic Label Embedding``, ``Hyperbolic Visual Embedding`` and ``Hyperbolic Gaussian Mixture VAE``, could the authors point out where they model the ``compositional labels`` related concepts?

---

> > ### Author Response · Authors · 2023-12-01
> > **reply**
> >
> > We apology for the confusion. This part is actually addressed in Sec. 4.1 [Hyperbolic Probabilistic Label Embedding], and we include additional explanations to address your concern.
> >
> > As presented, "we represent each label as a unimodal Gaussian in the hyperbolic space, and for each sample with multiple labels, its label embedding belongs to a Gaussian mixture subspace". This means that, for samples with multiple labels, such as "vehicle, large-vehicle, bus", the corresponding label embedding is crafted as a mixture of three Gaussian distribution, each aligning with one of the three categories.  The formal definition is given in Eq. 4:
> >
> > $p_\theta(z) = \frac{1}{\sum_{l=1}^Ly_l}\sum_{l=1}^L1(y_l=1)\mathcal{N}_p(z|\mu_l, \Sigma_l)$.
> >
> > Here the indicator function $1(y_l=1)$  determines the activation of each Gaussian distribution. For the above-mentioned example  with categories "vehicle, large-vehicle, bus", the Gaussian mixture can be specifically computed as:
> >
> > $p_\theta(z) = \frac{1}{3}(\mathcal{N}_p(z|\mu\_{{vehicle}}, \Sigma\_{vehicle}) + \mathcal{N}_p(z|\mu\_{large-vehicle}, \Sigma\_{large-vehicle}) + \mathcal{N}_p(z|\mu\_{bus}, \Sigma\_{bus}))$.
> >
> > This formulation illustrates how we compose multiple labels while ignoring the relationships among them. We understand that the writing can be enhanced, and will incorporate the discussions above into the upcoming version. Thanks.

---

> > > ### Comment · Reviewer_Lirp · 2023-12-02
> > >
> > > Thank you for the discussion we had. I have reviewed all the responses from our discussion again and will take them into consideration.

---

> > > > ### Author Response · Authors · 2023-12-02
> > > > **reply**
> > > >
> > > > We are grateful for your time and efforts engaged in the discussion. It is valuable for us to improve the manuscript.
> > > >
> > > > In addition, we'd like to take this final chance to update you the new results on "Probabilistic Label Embedding's effectiveness against non-probabilistic embeddings". For the baseline, we learn the embedding as a learnable vector, and change the alignment loss from KL loss to $\ell_1$ loss. At the moment, we only got the results for ImCLEF07A due to the time constraint, but will continue working on remaining datasets.
> > > >
> > > > |Variant | ImCLEF07A (mAP, CmAP, CV) |
> > > > |:----------|:----------|
> > > > | Baseline (Non-prob. embeddings)  | 86.14, 87.07, 8.44 |
> > > > | Our Full Model |  **92.21**, **92.44**, **3.18** |
> > > >
> > > > It is evident that our approach outperforms the baseline by solid margins across the three metrics. This verifies the efficacy of our design. The results will be incorporated to the article. Thanks.

---

> > > > > ### Comment · Reviewer_Lirp · 2023-12-04
> > > > >
> > > > > Thanks for the extra experiment. However, it does not support the claim ``Probabilistic Label Embedding's effectiveness against non-probabilistic embeddings``.
> > > > >
> > > > > I have to point out that in this experiment, the baseline is particularly weak and not suitable for supporting the claim because evaluating $\ell_1$ loss on a hyperbolic space seems to be meaningless and ill-defined.
> > > > >
> > > > > In fact, the performance gain shown in this extra experiment is likely (or at least possible) to be from **hyperbolic metric over euclidean metric** rather than **hyperbolic probabilistic over hyperbolic deterministic**.
> > > > >
> > > > > This specific experiment actually supports the claim that **hyperbolic metric (KL-loss based on Wrapped Normal) on hyperbolic geometry performs better than euclidean metric ($\ell_1$ loss) on hyperbolic geometry**.
> > > > >
> > > > > Also, $\ell_1$ is different from the proper baselines (Liu et al., 2020) and (Long et al., 2020) that I pointed out in previous discussions, where they use **deterministic hyperbolic embeddings with a hyperbolic metric**.

---

### Official Review · Reviewer_rCAE · 2023-11-19

**Soundness:** 3 good
**Presentation:** 2 fair
**Contribution:** 2 fair
**Rating:** 6
**Confidence:** 2

**Summary:**

The paper introduces a novel approach named HVSA (Hyperbolic Visual-Semantic Alignment) for the task of structural visual recognition. HVSA consists of two key components: hierarchy-agnostic visual-semantic alignment and hierarchy-aware semantic learning. Experimental results on various tasks and datasets demonstrate the effectiveness of the proposed method.

**Strengths:**

1. The paper provides a compelling motivation for incorporating the hierarchical nature of features into visual recognition tasks.

2. The authors perform comprehensive experiments across various tasks and datasets, yielding convincing results.

**Weaknesses:**

1. The paper primarily discusses the advantages of conducting feature learning in hyperbolic space compared to Euclidean space in a theoretical manner, highlighting the inherent exponential growth characteristic of hyperbolic embeddings in capturing hierarchical structures. However, there is a lack of experimental evidence to support this design choice. A potential baseline approach could involve performing all the loss terms in Euclidean space, such as triplet loss in Euclidean space.

2. The paper does not provide explicit details on how the taxonomy of labels in different datasets is obtained. It would be beneficial to include information regarding the acquisition of label taxonomy and potentially include tree structures to visually illustrate the hierarchical relationships within the taxonomy.

3. The experiment section lacks a brief introduction to the metric terms CV and CMAP.

**Questions:**

This is not a question regarding this paper's issue, but more like a open discussion. Since the paper demonstrates the benefits of incorporating taxonomy as prior knowledge for recognition tasks in a close-vocabulary setting. It raises an intriguing question about the scalability of this approach to an open-vocabulary setting. Considering the superior performance achieved by visual-semantic alignment methods like CLIP on in-the-wild data, I am wondering whether this work can provide similar advantages in the open-vocabulary domain.

**Details Of Ethics Concerns:**

No.

---

> ### Author Response · Authors · 2023-11-21
> **Point-to-Point Response to Reviewer rCAE**
>
> Thank you for your review. We have responded to your questions below, and wherever feasible, revised the paper to reflect these answers.
>
> To begin with, we focus on the following comment:
>
> > The paper primarily discusses the advantages of conducting feature learning in hyperbolic space compared to Euclidean space in a theoretical manner, highlighting the inherent exponential growth characteristic of hyperbolic embeddings in capturing hierarchical structures. However, there is a lack of experimental evidence to support this design choice. A potential baseline approach could involve performing all the loss terms in Euclidean space, such as triplet loss in Euclidean space.
>
> Sorry for the confusion.  We had already provided the experiments to examine the choice of latent space. Please refer to **Table 7** (copied below), in which `Hyperbolic` indicates our approach, while `Euclidean` means that all losses are computed in Euclidean space. The results show that using a hyperbolic space in our approach yields **consistently** and **notably** improvements against using a Euclidean space, across four datasets. This supports our theoretical analysis and evidences the efficacy of our algorithmic design. Thanks.
>
> |Geometric space | City (mIoU$^1$, mIoU$^2$) | PASCAL (mIoU$^1$, mIoU$^2$, mIoU$^3$)  | ImCLEF07A (mAP, CmAP, CV) | ImCLEF07D (mAP, CmAP, CV) |
> |:----------|:------------:|:------------:|:---------:|:------------:|
> | Euclidean  | 83.04, 92.41 | 74.17, 87.11, 96.42 | 88.47, 90.14, 5.47 | 87.61, 88.47, 6.98 |
> | Hyperbolic | **84.63**, **94.27** | **76.37**, **88.94**, **97.88** | **92.21**, **92.44**, **3.18** | **90.76**, **91.42**, **5.71** |
>
>
> Next, we address the two comments on writing:
>
> > The paper does not provide explicit details on how the taxonomy of labels in different datasets is obtained. It would be beneficial to include information regarding the acquisition of label taxonomy and potentially include tree structures to visually illustrate the hierarchical relationships within the taxonomy.
>
> Thanks for pointing this out.  All the hierarchical structures we used are officially defined in respective datasets. We directly use them without any modification. Following your suggestion, we have illustrated detailed semantic hierarchies on the datasets in Appendix Fig. 2 and Fig. 3. Thanks.
>
> > The experiment section lacks a brief introduction to the metric terms CV and CMAP.
>
> We had provided explanations to CV and CMAP in Appendix B.2. In the revised article, we additionally include formal definitions of the two terms (see Appendix Eq. 15 and Eq. 16). Thanks.
>
> We guess that the reviewer missed this part due to the lack of proper reference in the main text, and we have fixed this. Thanks.
>
>
> Further, we offer our thoughts to the open question:
>
> > This is not a question regarding this paper's issue, but more like a open discussion. Since the paper demonstrates the benefits of incorporating taxonomy as prior knowledge for recognition tasks in a close-vocabulary setting. It raises an intriguing question about the scalability of this approach to an open-vocabulary setting. Considering the superior performance achieved by visual-semantic alignment methods like CLIP on in-the-wild data, I am wondering whether this work can provide similar advantages in the open-vocabulary domain.
>
> This is an interesting question and our insights are in two aspects:
>
> First, theoretically, our algorithm can be promoted to open-vocabulary domain, since its basic idea is consistent with CLIP. However, in practice, there are challenges: 1) models like CLIP require very large-scale training data like image-text pairs, and explicit modelling the hierarchical structure within the data (as our method does) is hard if not impossible. Hence, implicit hierarchal structure learning appears to be more favorable and the proposed triplet loss that relies on pre-defined label taxonomy might be not suitable. 2) in comparison with contrastive-like loss used in CLIP, end-to-end optimization of VAE for large models is more difficult and expensive. Therefore, greedy optimization algorithms seem to be crucial as well.
>
> Second, though not closely relevant to your question, we highlight that our model can serve as a more trustworthy system to handle open-vocabulary cases, as compared to current recognition models. Consider a simple label taxonomy: `Animal` $\rightarrow$ `Okapi`. For an Okapi image, even we might be not sufficiently trained to identify rare animal `Okapi`, our model can still hold a high confidence to interpret the image belonging to a broader category of `Animal'.  This feature makes our algorithm fit to safety-critical applications (e.g., autonomous driving), even it itself is not purposely trained for open-vocabulary recognition.  Thanks.

---

> > ### Comment · Reviewer_rCAE · 2023-11-22
> > **Thank you and another question**
> >
> > Thank you for providing clarification. I have one more question that may go beyond the scope of this paper: In your discussion, you mentioned that "Hence, implicit hierarchical structure learning appears to be more favorable." Could you please provide further elaboration on how to perform implicit hierarchical structure learning? Are there any recommended literature on implicit hierarchical structure learning in open vocabulary that you can suggest?
> >
> > P.S. I have updated my scores as my concerns regarding this paper have been addressed.

---

> > > ### Author Response · Authors · 2023-11-22
> > > **Thank you for your response and further discussions.**
> > >
> > > We appreciate your engagement in the discussion and are pleased to see that you are happy to our rebuttal.
> > >
> > > Here we provide more explanations on "how to perform implicit hierarchical structure learning". The idea is actually very widely explored in pure vision tasks to capture (i) hierarchical relationships between objects/scenes [ref1,ref5], (ii) latent hierarchical structures naturally in vision datasets [ref2,ref3], and (iii) the structures inherent in video frames [ref4] or 3D medical data [ref6]. Techniques like hierarchical clustering or hyperbolic geometry, are typically used to discover or model the implicit structures.
> > >
> > > Moreover, similar ideas have also been studied for multimodal (image and language) learning, which might be more closely relevant to your question. For example, [ref7] explores the hierarchical relations between sentences (or image captions), phrases, images, and regions to learn structural visual-semantic representations. Strictly speaking, the method is not implicit, because it needs a step to parse sentences and images into explicit hierarchies, but the good thing is that it does not require a pre-defined taxonomy. [ref8] represents an extension of [ref7]. Beyond this, we'd like to highlight [ref9]. It builds a visual-semantic hierarchy, in which images are at the finest-grained level and its language descriptions are organized based on the semantic relationship. It then learns the partial order between sentences and images. Though this is an explicit hierarchical learning method, it should still work well in real-world, even open-vocabulary cases in which a complete taxonomy is not available, for example, enable neural networks to understand natural, more complex structures, by exploring the partial embedding order in a semantic hierarchy that is incomplete but easy to built.
> > >
> > > For the open-vocabulary domain, hierarchical structures have been confirmed to be crucial in, such as, [ref10,ref11]. However, most works are still explicit methods, and we did not identify closely matched implicit methods.
> > >
> > > Thanks again for your comments. Please let us know if you have any other concerns.
> > >
> > > ---------
> > > [ref1] Hierarchical Semantics of Objects (hSOs). ICCV 2007
> > >
> > > [ref2] Hyperbolic Image Embeddings. CVPR 2020
> > >
> > > [ref3] Hyperbolic Image Segmentation. CVPR 2022
> > >
> > > [ref4] Learning the Predictability of the Future. CVPR 2021
> > >
> > > [ref5] Unsupervised Discovery of the Long-Tail in Instance Segmentation Using Hierarchical Self-Supervision. CVPR 2021
> > >
> > > [ref6] Capturing implicit hierarchical structure in 3D biomedical images with self-supervised hyperbolic representations. NeurIPS 2021
> > >
> > > [ref7] Hierarchical Multimodal LSTM for Dense Visual-Semantic Embedding. ICCV 2017
> > >
> > > [ref8] Unified Visual-Semantic Embeddings: Bridging Vision and Language with Structured Meaning Representations. CVPR 2019
> > >
> > > [ref9] Order-Embeddings of Images and Language. ICLR 2016
> > >
> > > [ref10] PyramidCLIP: Hierarchical Feature Alignment for Vision-language Model Pretraining. NeurIPS 2022
> > >
> > > [ref11] Hierarchical Open-vocabulary Universal Image Segmentation. NeurIPS 2023.

---

### Author Response · Authors · 2023-11-21
**General Response**

We thank reviewers for their valuable feedback.

We are encouraged that the reviewers recognise the contributions of our work. Specifically, they find our paper _providing a compelling motivation_ (rCAE), our solution _novel_ (Lirp, rCAE), _reasonable and sound_ (SCte). Reviewer SCte remarks that "_the paper is well-written and easy to follow_". All reviewers confirm that our experiments are extensive, and results are _convinced_ (rCAE) and _competitive _(SCte).

In response to reviewers' comments and suggestions, we have revised our paper to hopefully address all of them, and point out main changes below for convenience.

1. In Introduction (Sec. 1), we added new descriptions to better explain the motivations of our approach.
2. We re-organized the original methodology part into two separate sections: Sec. 3 for preliminaries of hyperbolic learning and Sec. 4 only for methodology, to address the concern of reviewer Lirp.
3. In the new Sec. 4.1, we provided the missed details about the method. Moreover, we re-written most parts of Sec. 4.2 to make the algorithm clearer.
4. In Sec. 4.3, we provided a more comprehensive description about network architecture, to hopefully address the concern of reviewer Lirp.
5. In Sec. 5, we offered more in-depth discussions on experiments results, as suggested by reviewer Lirp.
6. We illustrated the label taxonomy of four datasets in Appendix Fig. 2 and Fig. 3, based on the suggestion of reviewer  rCAE.
7. We offered more formal definitions of the metrics (i.e., CV and CMAP) in Appendix Eq. 15 and Eq. 16, as suggested by reviewer rCAE.
8. We provided additional analysis to computation costs of the method in Sec. B.3, as per the suggestion of reviewer SCte.
9. All of the typos, minor omissions, and unclear explanations pointed out by reviewers have been fixed.

We marked major updates in red to make it easier to identify the changes. More details and information of other changes can be found in the respective author responses for each reviewer.

---

### Meta-Review · Area_Chair_Pn5V · 2023-12-07

**Metareview:**

The paper proposes an approach for structural visual recognition. The paper proposes a hierarchy-agnostic visual-semantic alignment and  a hierarchy-aware semantic learning, which jointly leading to the o hierarchical alignment of visual-semantic features. The paper received two positive reviews and one strong negative ratings. The positive reviews are based on the novel idea of the visual-semantic alignment from a probabilistic perspective in hyperbolic space. The negative reviews are based on the concerns: (1) The contributions are entangled and are not well ablated, and some perspectives that differentiate the work the most are not well illustrated, including compositional label and probabilistic embedding, (2) One important reference, Ganea et al., 2018, is not properly discussed and explained, especially to the potential contradiction between the entailment cone based method and the usage of eq(4) to model embeddings. After discussions, we recommend the rejection of the paper and highly encourage the authors to improve and revise the paper accordingly.

**Justification For Why Not Higher Score:**

Some ablations of crucial components and discussions with important references are missing.

**Justification For Why Not Lower Score:**

N/A

---

### Decision · Program_Chairs · 2024-01-16

Reject